# A Mammalian enhancer trap resource for discovering and manipulating neuronal cell types

Yasuyuki Shima[1], Ken Sugino[2], Chris Martin Hempel[1†], Masami Shima[1], Praveen Taneja[1], James B Bullis[1], Sonam Mehta[1], Carlos Lois[3], Sacha B Nelson[1]*

[1]Department of Biology and National Center for Behavioral Genomics, Brandeis University, Waltham, United States; [2]Janelia Research Campus, Howard Hughes Medical Institute, Ashburn, United States; [3]Division of Biology and Biological Engineering, Beckman Institute, California Institute of Technology, Pasadena, United States

**Abstract** There is a continuing need for driver strains to enable cell-type-specific manipulation in the nervous system. Each cell type expresses a unique set of genes, and recapitulating expression of marker genes by BAC transgenesis or knock-in has generated useful transgenic mouse lines. However, since genes are often expressed in many cell types, many of these lines have relatively broad expression patterns. We report an alternative transgenic approach capturing distal enhancers for more focused expression. We identified an enhancer trap probe often producing restricted reporter expression and developed efficient enhancer trap screening with the PiggyBac transposon. We established more than 200 lines and found many lines that label small subsets of neurons in brain substructures, including known and novel cell types. Images and other information about each line are available online (enhancertrap.bio.brandeis.edu).

*For correspondence: nelson@brandeis.edu

Present address: †Galenea Corporation, Wakefield, United States

## Introduction

The mammalian brain is likely comprised of thousands of distinct neuronal cell types. The ability to distinguish these cell types and to understand their roles in circuit activity and behavior is enhanced by an increasing variety of new genetic technologies in mice. Conditional transgenes like fluorescent reporters or alleles that sense or modify neuronal activity can be turned on in cells of interest through the use of 'driver' strains selectively expressing Cre recombinase or the tet transactivator (*Huang and Zeng, 2013*; *Luo et al., 2008*). Most techniques for producing these driver strains rely on recapitulating endogenous patterns of gene expression. However, selective expression patterns often depend both on elements within the proximal promoter, and on enhancers and other regulatory elements that can be located quite distally (*Visel et al., 2009*). Recapitulating endogenous expression requires either a knock-in approach (*Taniguchi et al., 2011*), or making transgenics from very large genomic fragments containing both the promoter and distal control elements (e.g. BAC transgenics (*Gong et al., 2003*; *2007*; *Yang et al., 1997*)).

One limitation of recapitulating endogenous expression patterns is that they are often broader than would be optimal for selective control. For example, the Pvalb-cre driver strain (*Hippenmeyer et al., 2005*) can be used to target Pvalb-positive fast-spiking interneurons in the neocortex; however, Pvalb is also expressed in cerebellum (Purkinje cells), dorsal root ganglia, thalamus, and many other brain structures, as well as in skeletal muscle. Even in the neocortex, Pvalb-positive cells consist of at least two distinct interneuron subtypes (basket cells and chandelier cells) and some layer 5 pyramidal neurons. Limitations on cell type specificity are common, since most genes

**eLife digest** Scientists can track and even alter the activity of different kinds of neurons, as well as the connections between neurons, by manipulating their genes. However, most genes are active in many different kinds of cells in many different places in the brain, making it difficult to track or target only a particular neuron or brain area.

Enhancers are sections of DNA that can regulate the activity of nearby genes so that they are only active in very specific cell types, and an "enhancer trap" is a genetic approach that essentially hijacks enhancers to express artificial genes in those same cell types. The technique relies on inserting a genetic marker, which can be easily tracked, into random locations in the genome. If this marker then interacts with an enhancer, it is activated and the effect of the enhancer on gene expression can be assessed.

This method has been used in fruit flies and fish to identify enhancers that specifically restrict gene expression to a small subset of cells. Now, Shima et al. show that enhancer traps can be used successfully in mammals too. The experiments produced over 200 different strains of mice, many with the fluorescent marker only in specific brain areas or in specific kinds of brain cells. Some of the types of brain cells uncovered by these experiments are new, and the labelling of specific brain cells and brain areas in different strains makes these mice a useful resource for future work. Furthermore, it will be relatively straightforward to produce many more strains of these mice, because it would simply involve crossbreeding mice. It is likely that some of these to-be-discovered strains will be useful tools for research as well.

are expressed in many different cell types throughout many different brain regions and tissues. Although combinatorial approaches can enhance specificity (*Madisen et al., 2015*), this comes at the cost of increasing the number of alleles that must be created and bred. Furthermore, this approach requires initial knowledge about co-expression patterns that may be lacking for some cell types.

Here, we take an alternative approach that relies on the fact that some minimal promoters can, when randomly inserted into the genome, interact with local enhancers and regulatory elements to produce patterns of expression that can be more restricted. This approach, termed enhancer detection or enhancer trapping, has a long history in *Drosophila* where it has been pursued primarily using the Gal4-UAS system (*Bellen et al., 1989*; *Brand and Perrimon, 1993*). More recently, this system and others have been used for enhancer trapping in zebrafish (*Balciunas et al., 2004*; *Scott et al., 2007*; *Urasaki et al., 2008*), but the approach has been less widely used in mice (though see *Gossler et al., 1989*; *Kothary et al., 1988*; *Soininen et al., 1992*; *Stanford et al., 2001*). A large-scale enhancer trap screen was performed using the SleepingBeauty transposon system (*Ruf et al., 2011*) but was focused on enhancers active during embryonic development, rather than those that regulate cell-type-specific expression in the adult. Kelsch et al. (*Kelsch et al., 2012*) conducted a mouse enhancer trap screen for transgenic animals with specific patterns of neural expression. Their lentiviral enhancer probe successfully generated transgenic lines with expression in neuronal subsets, however, the number of lines generated was small and most lines had expression in many cell types. Thus, this approach, while promising, has not yet reached its full potential, both in terms of specificity and in terms of the efficiency with which new lines can be generated.

Here, we report on an efficient enhancer trap screen to generate lines with specific expression patterns in the brain. First, using lentiviral transgenesis (*Lois et al., 2002*), we discovered a tet-transactivator-dependent enhancer probe capable of generating transgenic lines with highly restricted expression patterns. Next, we incorporated this tet-enhancer probe into the PiggyBac transposon system and developed a simple and efficient system for producing mouse lines with different PiggyBac insertion sites. The majority of these lines have brain expression and many have highly restricted expression patterns in known or novel neuronal cell types. Finally, a critical consideration in using the enhancer trap approach in the CNS of any species is the question of whether trapped neurons represent specific cell types or more random subsets of largely unrelated cells. To address this, we performed more detailed anatomical and physiological characterization in a subset of lines. These

experiments revealed that the neuronal populations are not random assortments of unrelated cells, but represent highly specific, previously recognized, as well as novel, neuronal cell types. In addition, quantitative comparison with a recently annotated collection of knock-in and BAC-cre driver strains revealed that expression is, on average, far more restricted in the enhancer trap lines. Hence enhancer trapping is a viable strategy for producing driver strains that complement those generated through other genetic approaches. This resource provides a platform for genetic control of a wide variety of neuronal cell types, as well as for discovering new subtypes of known neuronal cell types.

## Results

### Lentivirus transgenesis

Our initial enhancer trap screen employed lentiviral vectors because their highly efficient transduction of transgenes to the germ line minimized the number of injections needed to sample enough founders and their random single copy insertion permitted a broad survey of genomic sites (*Lois et al., 2002*) (see *Figure 1—figure supplement 1A* for transgenesis scheme). Our enhancer probe constructs employed the tet-off genetic driver system and incorporated a tet-responsive element (TRE; we used TREtight, the second-generation TRE) driving the fluorescent reporter mCitrine, so that we could examine expression patterns in driver lines without crossing to separate reporter lines. We initially tried constructs with the minimal promoter from the mouse heat-shock protein 1A (Hspa1a) gene (*Bevilacqua et al., 1995*, *Figure 1—figure supplement 2*). We also incorporated other promoter sequences that had been used to generate transgenic animals with neuronal subset expression and enhancer candidate sequence from evolutionarily conserved elements (*Visel et al., 2007*). We found a construct containing the minimal HSP promoter most efficiently generated lines with specific expression patterns in brain (28.8%, see *Table 1*) and see supplemental note and *Figure 1—figure supplement 2* for details of other constructs tried.

Throughout the rest of the paper, we use the admittedly imperfect term 'cell type' to refer to cell populations defined operationally as the group of neurons labeled in a particular brain region of a transgenic line. We imagine neuronal cell types as nodes in a hierarchical tree-like structure with the terminal branches ('leaves') corresponding to 'atomic' cell types which are homogeneous and cannot be further divided based on projections, morphology, gene expression etc. The 'operational' cell types defined here are not necessarily 'atomic' in that further characterization may reveal that they are composed of subtypes, but they offer a useful starting point for subsequent identification of 'atomic cell types' based on uniformity of morphology, connections, physiology, and gene expression.

Although only a minority of lentiviral tet lines had reporter expression, the majority of lines with brain expression had highly restricted expression patterns. Some lines had expression only in restricted cell types, including medial prefrontal cortex layer 5 neurons (*Figure 1A*), retinal ganglion cells projecting axons to superior colliculus (*Figure 1B*), and Cajal-Retzius cells

**Table 1.** Efficiency of transgenesis. The numbers of lines dissected and the number of lines with brain expression are shown separately for each construct used.

| construct | analyzed | with expression |
|---|---|---|
| **lentivirus** | | |
| *HSP-tet* | | |
| HSP-tet | 132 | 38 |
| hsp-tet2 | 7 | 4 |
| hsp-tet3 | 12 | 0 |
| *promoter-tet* | | |
| CamKII-tet | 18 | 4 |
| minCamKN-te | 17 | 7 |
| Thy1-tet | 15 | 0 |
| minThy1-tet | 11 | 0 |
| Gad1-tet | 9 | 0 |
| Slc-tet | 6 | 0 |
| *enhancer-tet* | | |
| 119-tet | 13 | 1 |
| 121-tet | 2 | 0 |
| 122-tet | 15 | 2 |
| 170-tet | 2 | 1 |
| **PiggyBac** | | |
| *HSP-tet* | | |
| tet | 101 | 81 |
| tet-Cre | 57 | 43 |

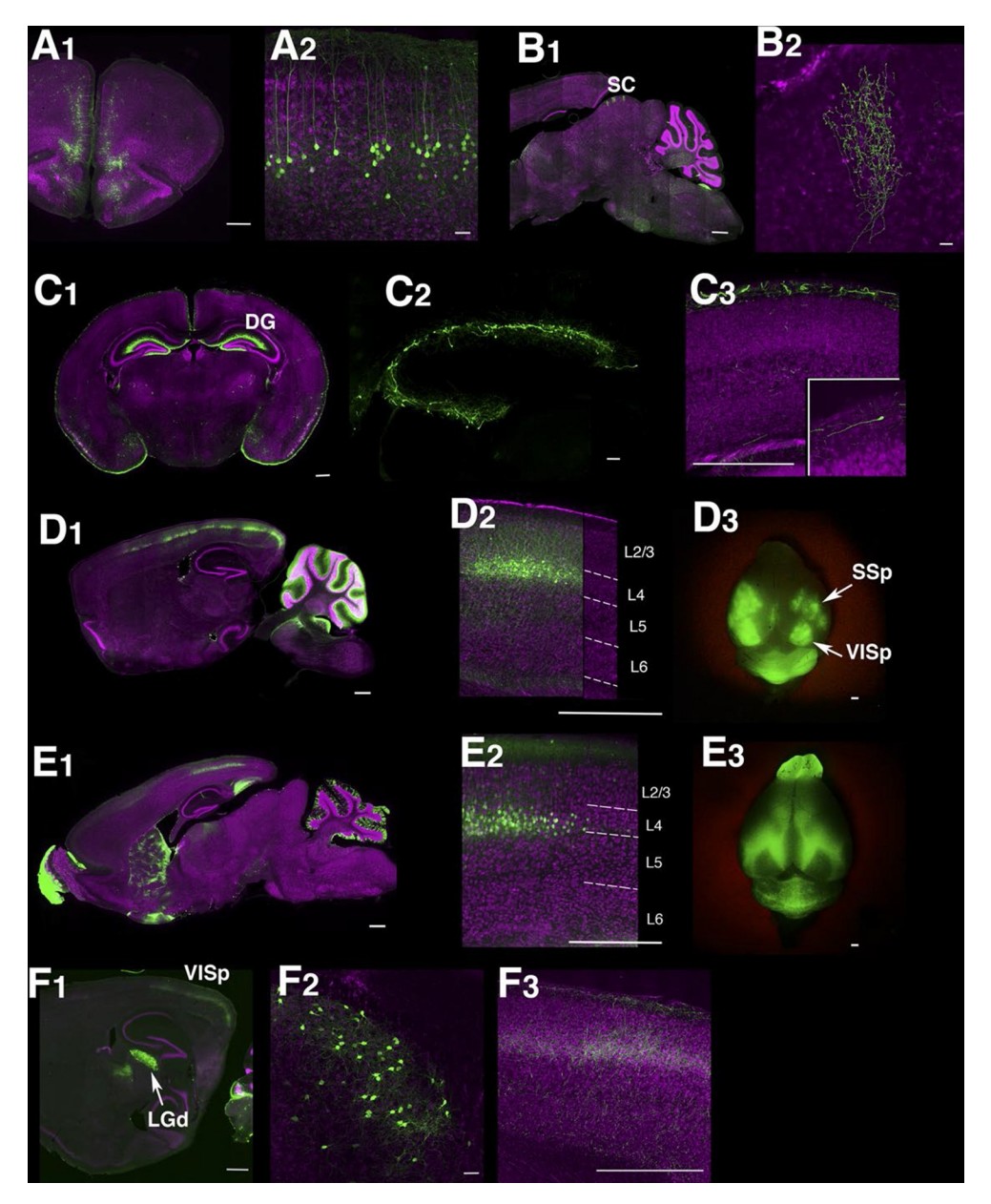

**Figure 1.** Example Lentiviral lines. (**A**) 48L has expression in limbic cortex (A1, coronal section) layer 5 pyramidal cells (A2, magnified image in limbic cortex). (**B**) Superior colliculus (SC) of TCBV has columnar axons from retina. B1: sagittal section, B2: magnified image of superior colliculus. (**C**) 52L has expression in piriform cortex (see *Figure 9*) and Cajal-Retzius cells in dentate gyrus (DG, C2) and cerebral cortex (C3, inset: magnified image of a Cajal-Retzius cell). (**D**) TCGS has expression in layer 4 neurons of primary sensory cortices (primary somatosensory area: SSp and primary visual area:VISp in D3). (**E**) TCFQ has nearly complimentary layer 4 expression excluding primary sensory cortices. D1 and E1: sagittal sections, D2 and E2: confocal images of cortex, D3 and E3: dorsal view of whole brains. (**F**) TCJD has expression in dorsal part of lateral geniculate nucleus (LGd, F1), which projects to primary visual cortex (VISp). F1: sagittal section, F2: higher magnification of LGd, F3: higher magnification of axons in layers 1, 4, and 6 of VISp. Scale bars are 50 μm in A2, B2, C2, F2 and 500 μm in others.

The following figure supplements are available for figure 1:

**Figure supplement 1.** Transgenesis.

**Figure supplement 2.** Constructs for transgenesis.

**Figure supplement 3.** Transgene regulation by Doxycycline (Dox).

in cerebral cortex and dentate gyrus (*Figure 1C*). We had two lines with distinctive expression in cortical layer 4 neurons; TCGS in primary sensory cortices (including primary visual, somatosensory and auditory cortices; *Figure 1D*) and TCFQ which was devoid of expression in primary sensory cortices but expressed in associative cortices (*Figure 1E*). We also obtained lines labeling specific cell types, such as thalamocortical projection neurons in the dorsal part of the lateral geniculate complex (LGd; *Figure 1F*), anatomically clustered subsets of cerebellar granule cells, semilunar cells and a subset of superficial pyramidal neurons in piriform cortex, and a subtype of cortico-thalamic pyramidal neurons in layer 6 of neocortex (see below). Tet reporter expression could be turned off and on by administration of doxycycline (*Figure 1—figure supplement 3*). For a summary of expression patterns in all lines, see *Supplementary file 1*.

Although lentiviral transgenesis successfully generated lines with highly restricted expression patterns, screening was difficult to scale up effectively. Generation of each new founder requires viral injection into single cell embryos and transfer of that embryo to a foster mother (*Figure 1—figure supplement 1A*). Tracking and segregating multiple alleles in order to identify the allele responsible for reporter expression in the case of founders having multiple insertions was especially time consuming. Moreover, we found four lines in which expression in the founder and early progeny was lost in later generations, implying possible silencing of lentiviral transgenes over generations (*Hofmann et al., 2006*). We tried to incorporate insulator sequences (see below) to prevent silencing, but viruses with insulator sequences had 100 times lower titer (about 1 x 10$^7$ infection unit/ml) and were not usable for transgenesis.

## PiggyBac transgenesis

In order to develop a more efficient and scalable transgenesis platform, we made use of the PiggyBac (PB) transposon system as a means of delivering tet enhancer trap probes. The PB system has been widely used in mammalian genetics (*Ding et al., 2005*) for insertional mutagenesis (*Rad et al., 2010*) and stable transgene expression (*Woodard and Wilson, 2015*). Unlike the SleepingBeauty transposon, PiggyBac has a weaker tendency to undergo local hops (*Liang et al., 2009*, but see supplemental note), making it more suitable for screens that target the whole genome. To simplify the process of establishing and tracking new transgenic alleles, we established lines of animals carrying a single-copy PB integration and additional lines expressing the PiggyBac transposase (PBase). PB hops only in animals with both the PB and PBase alleles, allowing us to generate transgenic animals with different PB insertion sites simply by mating wild type and PB;PBase animals (see the mating scheme in *Figure 1—figure supplement 1B*). We used the same tTA-reporter system used in the lentiviral probes (PB tet, *Figure 1—figure supplement 2B*). We also created a probe designed to produce both tTA and Cre expression (PB tet-cre, *Figure 1—figure supplement 2B*). In order to prevent the silencing seen in some lentiviral lines, we incorporated barrier insulator sequences from the chicken β-globin gene (see 'Materials and methods' for detail).

DNA plasmids encoding our PB enhancer probe and mRNA encoding a hyperactive PBase (*Yusa et al., 2011*) were co-injected into single cell embryos. Among 28 PB-positive animals, five had single-copy insertions confirmed by quantitative PCR and ligation-mediated PCR. These five served as seed lines for subsequent rounds of piggyBac transgenesis.

We used two PBase lines: 1) a Rosa-PBase line generated by Allan Bradley and colleagues (*Rad et al., 2010*) having nearly ubiquitous expression of PBase from the Rosa-26 locus and 2) a Prm-PBase line that we generated having spermatid-specific expression of hyper-active PBase (*Yusa et al., 2011*) under the protamine-1 promoter (*Zambrowicz et al., 1993*). PiggyBac seed lines were crossed with PBase mice, and PB;PBase double hemizygous animals (P1 generation) were selected and crossed with wild-type animals (see *Figure 1—figure supplement 1B*). P2 generation animals were genotyped for PB alleles (*Table 2*).

Animals carrying the PB allele were further tested to ensure transposition had occurred. We found that the PB allele transmission rate was significantly lower than the expected Mendelian ratio, implying that a substantial fraction of excised PB failed to re-insert into the host genome (*Table 2*). The PB alleles derived from each of the single-copy seed lines jumped at similar transposition rates, except for those from the PBAQ seed line that rarely translocated (*Table 2*). We found that Prm-PBase produced founders more efficiently than Rosa-PBase (*Table 2*). See supplemental note and *Tables 3* and *4* for further details of transposition frequency.

**Table 2.** Transposition efficiency PB;PBase double hemizygous animals (PB/+; PBase/+) were crossed with wild type animals and genotypes of pups from the mating were examined (see the mating scheme in *Figure 1—figure supplement 1B*). Numbers of animals are shown in parentheses. PB transmission rate: number of PB+ animals / total number of animals, PB transposition rate: number of PB in new sites / number of animals tested for transposition. (Note: we did not test transposition for PB/+;Rosa-PBase/+ and PB/+;Prm1-PBase/+ males because transgenes might not be stably transmitted to the next generation in these animals). New line production efficiency: number of animals with new insertion site / total number of animals born. *: All PB+ animals were female.

| Seed line | PBase line | PB transmission rate | Transposition efficiency | Efficiency of new line production |
|---|---|---|---|---|
| PBAG | Rosa | 28.6% (54/189) | 41.4 % (12/29) | 6.4 % (12/189) |
|  | Prm1 | 29.2% (21/72) | 56.3 % (9/16) | 12.5% ( 9/72) |
| PBAW | Rosa | 21.56% (80/371) | 62.2 % (23/37) | 6.2 % (23/371) |
|  | Prm1 | 33.0 % (97/294) * | 67.4 % (62/92) | 21.1% ( 62/294) |
| PBAS | Rosa | 30.8 % (33/107) | 25.0 % (3/12) | 3.3 % ( 3/97) |
|  | Prm1 | 35.6 % (130/365) | 41.9 % (39/93) | 10.7 % (39/365) |
| PBAU | Rosa | 22.2 % (30/135) | 38.9 % (7/18) | 5.2 % (7/135) |
|  | Prm1 | 34.3 % (46/134) | 57.1 % (20/35) | 14.9 % (20/134) |
| PBAQ | Rosa | 37.5 % (6/16) | 0 % (0/3) | 0 % (0/16) |
|  | Prm1 | 60.0 % (9/15) | 12.5 %( 1/8) | 6.6 % (1/15) |

## PB line expression patterns

We established more than 200 independent lines and examined expression patterns from more than 130 lines (210 and 135 as of October 2015, respectively; see *Supplementary file 1* for line expression summary and insertion sites). We occasionally encountered termination of lines for infertility (20 lines; cf. the productive mating rate of the mouse strain C57Bl6/j is 84% [*Silver, 1995*]) or death mainly due to maternal complications at birth (5 lines). Because of the difficulties associated with managing a large number of colonies, some (4 lines) were accidentally lost before cryopreservation.

The rate of obtaining lines with brain expression in the PB screen (78.5% ) was more than twice that obtained with lentiviral transgenesis (*Table 1*). Lines generated by local hop of PB (within ∼100 Kb) had similar expression patterns to that of the original line (*Figure 2—figure supplement 1*), probably because shared local enhancers regulated expression of the reporter. In most of lines, we did not find clear resemblance between reporter expression patterns and those of genes near insertion sites (see supplemental note). Some lines had dominant expression in a single anatomical structure, such as deep entorhinal cortex (P038, *Figure 2A*), subiculum (P141, *Figure 2B*), retrosplenial cortex (P099, *Figure 2C*), or dorsal hindbrain (P108, *Figure 2D*). Many lines had expression in multiple regions but with unique cell types in each area. For example, P008 has broad expression in striatum (*Figure 2E1*) but has restricted expression in the most medial part of the hippocampus (fasciola cinereum, *Figure 2E2*). P057 had cortical layer 5 expression and restricted expression in anterior-lateral caudate putamen (CP; *Figure 2H*). Interestingly, the mCitrine-positive CP cells appeared to be part of the direct pathway; the cells projected axons to a limited area in substantia nigra pars reticulata (*Figure 2H2* inset) but not to the globus pallidus (*Figure 2H1* arrow; compare the GP projection of P008 in *Figure 2E1*). Lines with broad expression, (*Figure 2J*), those labeling few cells, and those closely resembling existing lines were terminated (48 lines). Most lines with 'broad expression' had strong mCitrine expression restricted to forebrain and founders carrying multiple PB copies also had strong forebrain expression.

We compared reporter expression patterns with those observed in BAC-Cre and knock in –Cre lines. Harris and her colleagues (*Harris et al., 2014*) manually evaluated the density of reporter-

**Table 3.** Numbers of insertion events occurring in genes and intergenic regions.

| Insertion Sites | Number of lines |
|---|---|
| gene | 60 |
| coding exon | 1 |
| 3'UTR | 4 |
| intron | 55 |
| intergenic | 81 |
| repetitive sequence | 26 |

positive cells in 295 brain structures for each of 135 BAC- and knock in-Cre lines into six categories (widespread, scattered, sparse, enriched, restricted/Laminar, and restricted but sparse). We employed the same expression categories to annotate expression patterns in PB lines (see 'Materials and methods' and *Figure 3—figure supplement 1*; annotation data is summarized in *Supplementary file 2*). We found that, on average, more than three times fewer structures were labeled in PB lines (33) than in the Cre lines (107) (*Figure 3A and B*). The numbers of structures with enriched/restricted expressions were also lower in PB lines (*Figure 3C*). We count the number of lines with expression in 12 major subclass of brain structures (*Harris et al., 2014*). Cre lines had relatively homogenous expression rates (70–83% ) in any brain regions, whereas PB lines had expression bias to forebrain structures such as the isocortex, olfactory bulb, and hippocampus (*Figure 3D*).

Although our screen was focused on brain expression, we also performed a brief screen of the rest of the body and found that some lines (24/135) also had expression in tissues other than the brain, including skin (8), bone (5), viscera (9), and brown (1) and white (3) adipocytes. Although we occasionally observed expression, in retinal ganglion cells (*Figure 1B*), and spinal cord (P032, P105), we did not systematically examine the retina, spinal cord, or peripheral nervous system. Non-brain expression patterns are summarized at the enhancer trap web site (enhancertrap.bio.brandeis.edu/data/).

## Expression stability in PB lines

Except for lines that were lost or terminated early, we examined expression patterns of multiple animals from each transgenic line (73 lines). Nearly all had consistent expression patterns over multiple generations. A few lines showed variable expression patterns in individual animals. P039 had stable expression in subiculum, but its expression in cortex varied, and two lines (P027 and P197) had expression that was left-right asymmetric. P139 heterozygous animals had consistent expression in cortico-thalamic L6 cells in lateral cerebral cortex (see *Figure 10*), but the number of labeled cells varied across animals and sometimes across hemispheres (data not shown). Since P139 homozygous animals had stable expression patterns, subthreshold-level tTA expression might have caused stochastic reporter expression.

In the brain, most enhancers display developmental dynamics visible, for example, in the state of an active enhancer marker H3K27Ace (*Nord et al., 2013*; *2015*). Many lines showed different reporter expression patterns at different ages, likely reflecting the developmental dynamics of the trapped enhancers. Screening was primarily carried out in young adults (P20-30). We examined adult (P50 or later) expression in 34 lines. Mature expression was reduced in 12 of these lines, but was retained in the remaining 22 lines. Some lines showed complex spatiotemporal expression patterns. For example, young postnatal animals (P10-12) from line P162 had expression in primary sensory

**Table 4.** Rates of inter-chromosomal, intrachromosomal and local (within 2 Mb) transposition events. Some insertions were not located due to insertion in repetitive sequences.

| line | number of lines | inter-chromosomal hop | intra-chromosomal hop | local (<2 Mb) hop | not located |
|---|---|---|---|---|---|
| PBAW | 69 | 46 (66.7 %) | 11(15.9 %) | 8(11.6 %) | 12 (17.4 %) |
| PBAS | 46 | 26 (56.5 %) | 18 (39.1%) | 9 (19.6%) | 2 (4.3 %) |
| PBAU | 26 | 13 (50.0 %) | 8 (30.8 %) | 2 (7.7%) | 5 (19.2 %) |
| Total | 141 | 85 (60.2 %) | 37 (26.2 %) | 19 (13.4%) | 19 (13.4 %) |

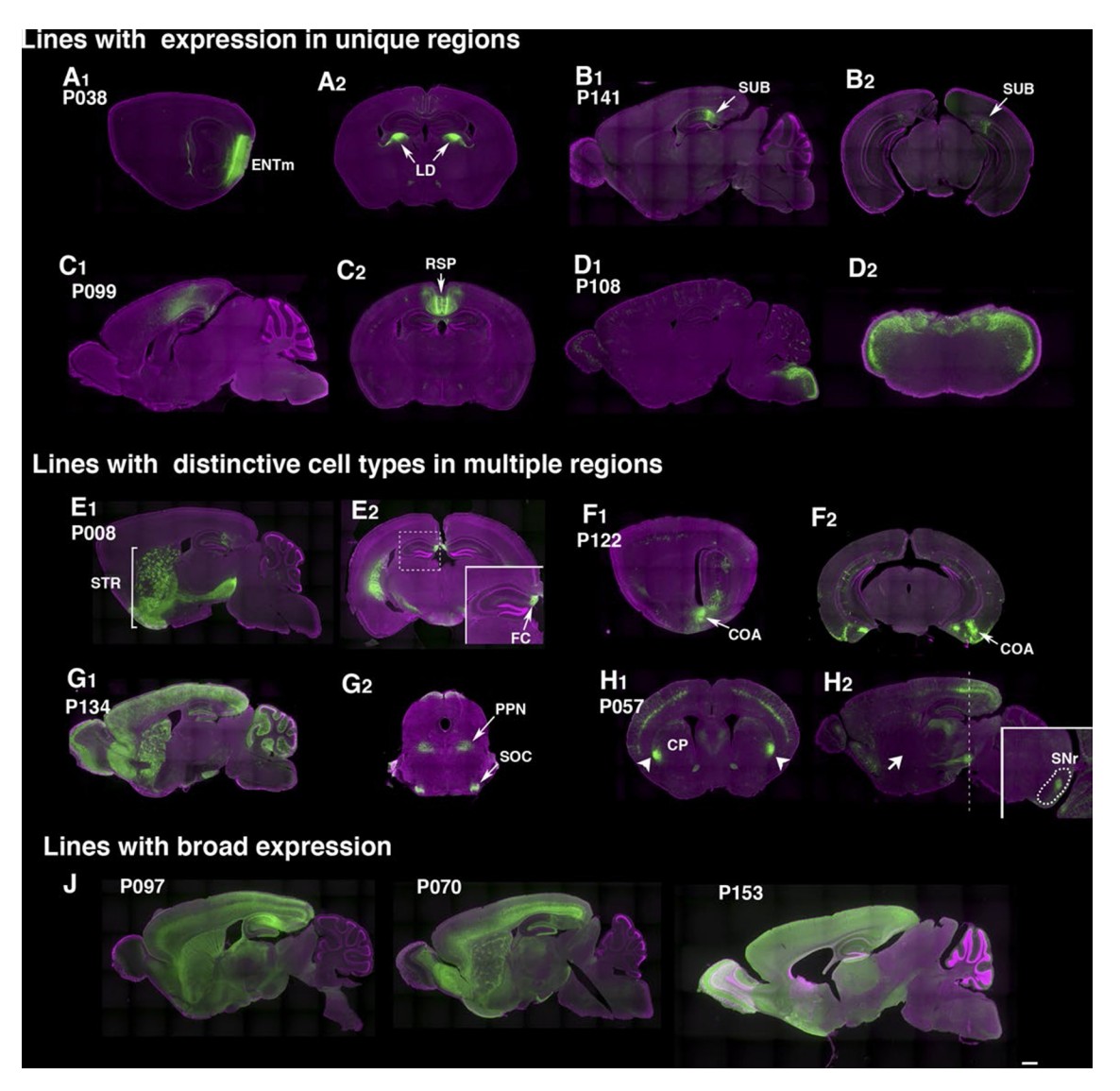

**Figure 2.** Example PiggyBac lines. (A–D) Examples of lines that appear to label a single cell type. (d) P038 has expression in entorhinal cortex medial part (ENTm) layer 6 neurons (A1: sagittal) that send axons to lateral dorsal nucleus of thalamus (LD in A2: coronal). (B) P141 has expression in a restricted area in subiculum (SUB, B1: sagittal, B2: coronal). (C) Retrosplenial cortex (RSP) expression in P099 (C1: sagittal, C2: coronal). (D) Dorsal hindbrain expression in P108 (D1: sagittal, D2: coronal at hindbrain). (E–H) Examples of lines with regionally distinctive cell type labeling. (E) P008 has expression in striatum (STR) broadly (E1: sagittal) but its hippocampal expression is restricted to the most medial part (fasciola cinereum: FC, E2 inset) (F) P122 has scattered expression in hippocampus and strong expression in cortical amygdalar area. F1: sagittal, F2: coronal sections. (G) P134 has broad expression in cortical interneurons and cerebellar Lugaro cells (G1: sagittal). Its expression in midbrain is restricted to subnuclei (G2, superior olivary complex: SOC and presumably pedunculopontine nucleus: PPN). (H) P057 (H1:coronal, H2, sagittal section) has expression in layer 5 pyramidal cells in the cortex. Expression in caudate putamen (CP) is restricted to lateral-most areas (arrows in H1). H2 inset: coronal section at the level of the dotted line. The striatal neurons project to a small area in the reticular part of the substantia nigra, reticular part (SNr, dotted area in H2 inset) but not to globus pallidus (H2 arrow). (J) Lines with broad expressions. Scale bar: 500 µm.

The following source data and figure supplements are available for figure 2:

**Source data 1.** Viral reporter expression counting data One or two animals per line were injected with TRE3G –myristorylsted mCherry HA.
**Figure supplement 1.** Similar expression patterns in lines with nearby insertions.
**Figure supplement 2.** Developmental dynamics in P162 expression patterns.

*Figure 2 continued on next page*

*Figure 2 continued*

**Figure supplement 3.** Examples of virus injection.
**Figure supplement 4.** tet reporter expression in cultured cell lines.
**Figure supplement 5.** Specificity of tet reporter expression in vivo.

cortices, the parafascicular nucleus of the thalamus and pontine grey, but more mature (3 weeks or older) animals lost thalamic and pontine expression and gained subiculum expression (*Figure 2—figure supplement 2*), suggesting the probe trapped enhancer(s) activated in different structures at different developmental stages. Unlike some lentiviral lines, silencing of PB transgenes over generations was not observed; even lines losing expression late in adulthood had pups that regained reporter expression.

## Transgene expression in tet lines

We examined whether our tet lines could drive transgenes other than the mCitrine encoded in the probes. TRE promoters are known to have weak, tTA-independent ('leak') expression that can be substantial when many copies of the TRE-Cre constructs are delivered virally (*Mizuno et al., 2014*; *Zhu et al., 2007*). We injected adenoassociated virus (AAV) encoding TRE-driven transgenes into

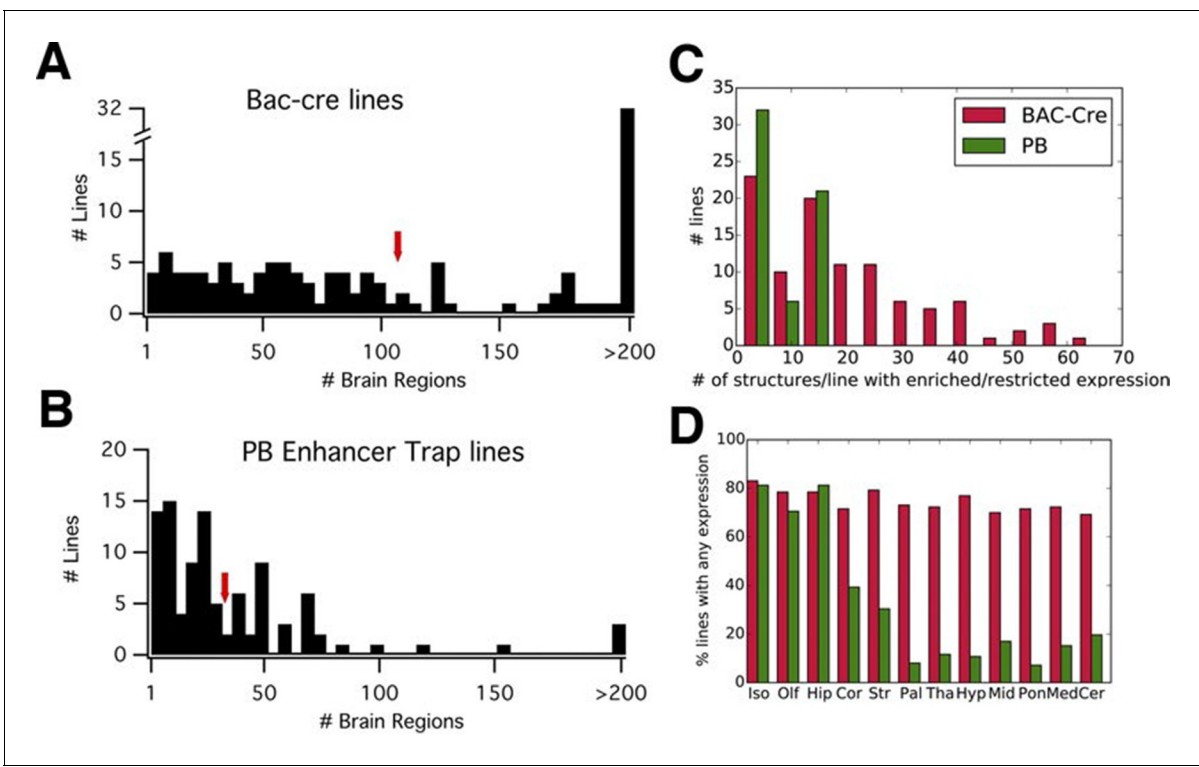

**Figure 3.** PB lines have more restricted expression than Cre lines. (**A**, **B**) Histograms of the number of brain regions (x axis) with expression per line. Bac-Cre/Cre knock in –lines (**A**) have expression in more areas than PB enhancer trap lines (**B**). Arrows: averages. (**C**) Histogram of number of brain structures with enriched or restricted expression. Red: Cre lines, Green: PB lines. (**D**) Fraction of lines with expression in brain subregions. Iso: Isocortex, Olf: olfactory areas, Hip: hippocampal formation, Cor: cortical subplate, Str: striatum, Pal: pallidum, Tha: thal amus, Hyp: hypothalamus, Mid: midbrain, Pon: Pons, Med: medulla, Cer: cerebellum. Red: Cre lines, Green: PB lines.

The following figure supplement is available for figure 3:

**Figure supplement 1.** Categories of regional expression patterns.

multiple areas in different lines (*Figure 2—figure supplement 3* and *Figure 10*. See *Figure 2—source data 1* for counts). We found mCherry reporter was expressed specifically expressed in mCitrine-positive cells in most of cases. For example, three lines with expression in different layers of retrosplenial cortex had specific virus expression in different layers (*Figure 2—figure supplement 3A–C*). Infection efficiency varied from 36.5% (CA1 pyramidal cells in P066, *Figure 2—figure supplement 4D*) to nearly 100% (layer 6 pyramidal cells in 56L, *Figure 10R and T*), probably due to AAV serotype preference. In some lines, as reported (*Choy, 2015*), we found a few viral reporter-positive but mCitrine-negative cells in the same layer/positions with those of mCitrine-positive cells (arrowheads in *Figure 2—figure supplement 3E and F*; 0.7% +/- 0.7% of infected cells, n=6 injections in 5 strains), but never in ectopic positions lacking mCitrine positive cells. We speculate this 'off-target' expression might be a result of competition over tTA proteins between single copy TRE of mCitrine in the genome and many copies of TRE from virus. Myristoylated mCherry driven by a second-generation TRE (TREtight-myrmCherry-HA) was expressed only in mCitrine-positive cells and could be used to map their axonal projections (see below). Similarly, tet-dependent channelrhodopsin virus (TREtight-ChR2H134R-mCherry) had specific expression only in mCitrine-positive cells and could drive action potentials upon blue light stimulation (*Choy, 2015*).

Because of the widespread utility of recombinase systems such as Cre and Flp, we made significant efforts to make reagents allowing either a) TRE-dependent recombinase expression or b) expression of Cre directly from the enhancer probe. Nearly all these attempts were unsuccessful (see *Figure 2—figure supplement 4* and *5*) due first to low level leak of all versions of the TRE promoter tried, combined with the high sensitivity of cre-dependent recombination. Expression of Cre from the enhancer probe may have suffered from this problem in some cases as well as the additional problem of more widespread developmental expression. We were able to obtain specific Cre-reporter expression restricted to mCitrine-positive cells, using an implementation of the GFP nanobody-split Cre virus (developed independently from *Tang et al., 2015*). The GFP nanobody-split Cre also had specific reporter expression from Ai14 (*Madisen et al., 2010*) TdTomato Cre reporter allele (supplemental note, *Figure 2—figure supplement 4* and *5*).

## A catalog of neuronal cell types

By screening a large number of lines, we were able to identify strains that target both classically distinguished neuronal cell types and subtypes of these cell types including some previously unrecognized subtypes. In this section, we focus on seven major brain structures. Our anatomical and physiological characterizations are necessarily incomplete, but we expect that others with scientific interests in the relevant structures will contribute to more detailed characterization.

### Neocortex

Many lines have regional and/or laminar expression in the neocortex. For examples, P172 (*Figure 4A*) had layer 4 expression in three primary sensory cortices (primary somatosensory area: SSp, primary visual area: VISp, and primary auditory area: AUDp) while P063 (*Figure 4B*), and P160 (*Figure 4C*) has expression in only in AUDp and VISp layer 4 neurons respectively. Layer 5 pyramidal neurons are categorized by their axonal projection patterns and somatodendritic morphologies (*Hattox and Nelson, 2007*; *Molyneaux et al., 2007*): callosal neurons (projecting to contralateral cortex) have thinner apical dendrites and smaller somata, whereas corticofugal (subcereberal) projection neurons have thick tufted dendrites and larger somata. Both main types of L5 neurons also project to the striatum. Corticofugal cells can be further divided based on their projection targets. P136 (*Figure 4D*) has expression in callosal thin tufted pyramidal neurons located in upper layer 5 (layer 5a) and projecting to the entire caudoputamen. P161 (*Figure 4E*) has expression in callosal projecting thin-tufted cells forming a thin layer above layer 6. P074 (*Figure 4F*) has expression in callosal projecting pyramidal neurons in anterior (motor and somatosensory) cortex with axonal projections that are biased to lateral caudoputamen. Thick tufted cortico-tectal neuronal lines (P081 and P084, *Figure 4G and H*) and cortico-spinal lines (P149 and P135, *Figure 4I and J*) also showed differences in regional expression patterns (P081: strong in somatomotor (MO) and supplemental somatomoter area (SSs), P084: only in SSp, P149: strong in MO and SSs, P135: SSp and VISp). We also obtained multiple layer 6 lines with different thalamic projection patterns (discussed below).

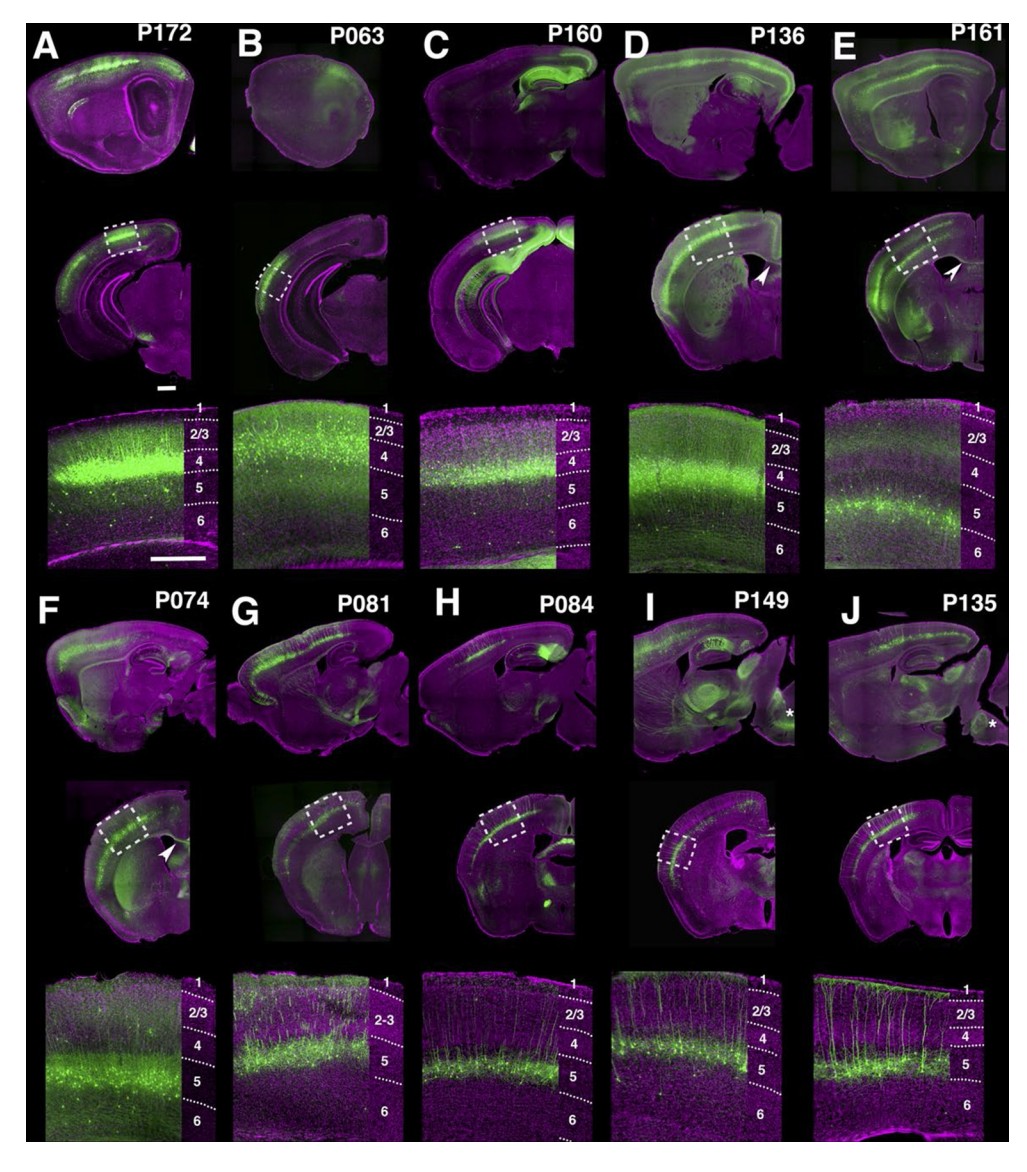

**Figure 4.** PB Lines labeling neocortical cell types. (A–J) Images of lines with layer-specific expression. Sagittal (top rows), coronal (middle rows), and high-magnification coronal images (bottom rows) are shown. Arrowheads (D,E"): callosal projections. Asterisks (I,J"): corticospinal projections. Scale bars: 500 µm.

The following figure supplement is available for figure 4:

**Figure supplement 1.** Retrosplenial cortex lines.

Layer-specific expression patterns were distinctive in retrosplenial cortex (RSP, *Figure 4—figure supplement 1*). P160, P099, and P136 have expression in distinctive layers of ventral part (RSPv) but not in dorsal part (RSPd). P160 has expression in layer 2 (*Figure 4—figure supplement 1A*), in P099 mCitrine cells form a thin layer immediately under layer 2 (*Figure 4—figure supplement 1B*), and P136 has expression in layer 2/3. P099 and P136 are strikingly different in their presence or absence of callosal projection (arrowhead). We also observed several lines with RSP layer 6 expression (*Figure 4—figure supplement 1D–F*), RSPd expression (P012, *Figure 4—figure supplement 1G*) and posterior RSPv expression (P122, *Figure 4—figure supplement 1H*).

## Olfactory bulb and related structures

The main olfactory bulb is a laminated structure that contains multiple cell types with different morphologies in each layer (*Nagayama et al., 2014*). P152 and P110 have expression in the glomerular layer of the main olfactory bulb (MOBgl). mCitrine-positive cells' dendrites point toward the centers of glomeruli in P152 (*Figure 5A*), but dendrites are mainly confined to glomerulus walls in P110 (*Figure 5B*). These morphological differences are characteristic of periglomerular cells and superficial short axon cells, respectively (*Nagayama et al., 2014*). There are several lines with expression in the outer plexiform layer (MOBopl). Many of these lines have an even distribution of labeled cell bodies within the MOBopl (P118 *Figure 5C*) but P157 (*Figure 5D*) has mCitrine-positive cells confined to the basal half of the layer. We also found lines targeting granule cells in the main olfactory bulb (MOBgr, P074, *Figure 5E*) and accessory olfactory bulb (AOBgr, P099, *Figure 5F*). We found lines with expression in the anterior olfactory nucleus. P135 has mCitrine-positive cell bodies in the outer layer (anterior olfactory nucleus layer 1, *Figure 5G*), whereas those of P113 are located in the ventral inner layer (*Figure 5H*). P074 has widespread expression in the inner layer (*Figure 5I*). We obtained three subtypes of piriform cortex layer II neurons (see below) and lines with expression in the cortical amygdalar area (COA, P055, and P122, see *Figure 2F*). We also had lines with expression in tenia tecta (ex. P064, data not shown).

## Hippocampal formation

The entorhinal cortex, hippocampus, and subiculum are interconnected by complex loops with reciprocal connections (*Ding, 2013*; *Witter et al., 2014*). Many (92) lines have expression in subregions of the hippocampal formation. CA1 is one of major input source of the subiculum, which sends axons to the entorhinal cortex through the presubiculum (PRE) and parasubiculum (PAR). Labeled cells proximal to CA1 in three lines, P162 (*Figure 6A*), P139 (*Figure 6B*), and P141 (*Figure 6C*), do not project to PRE but the distal subiculum population labeled in P157 (*Figure 6D*) does. P066, which has expression in the whole subiculum, also has PRE projections (*Figure 6E*). P162, P139, and P141 have expression in adjacent positions (cells in P162 and P139 are in nearly the same positions and P141 cells are located posterior to them, (see distal ends of CA1 marked by arrowheads in *Figure 6A–C*) but have different axonal projections. P162 has dense mCitrine-positive axons in the reuniens nucleus (RE) and dorsal and ventral submedial nucleus (SMT) but there are few axons in the corresponding areas in P139 and P141 (*Figure 6—figure supplement 1A–F*). Injection of tet – dependent virus (AAV-Tre3G-myristylated mCherry-HA) into the subiculum in P162 confirmed that axons in RE and SMT were coming from the subiculum (*Figure 6—figure supplement 1G–I*). P160 has expression in PRE and PAR and dense axonal projections to entorhinal cortex, medial part (ENTm, *Figure 6E*). P149 has expression in ENTm layer 5 (*Figure 6H*). P084 also has expression in the same region, but the expression is restricted to the most medial part of ENTm (*Figure 6G*). 56L had broad expression in deep layer 6 of neocortex, but its expression in ENTm was observed in upper layer 6 (*Figure 6J*). P038 occupied ENTm deep layer 6 (*Figure 6I*). We also obtained lines with ENTm layer 2/3 (PBAS, *Figure 6K*) and entorhinal cortex, lateral part (ENTl) layer 2/3 (P126, *Figure 6L*).

## Cortical subplate (claustrum, endopiriform, amygdala) and nuclei (striatum and pallidum)

We identified lines separately labeling endopiriform nucleus (P138, *Figure 7A*) and claustrum (P018, *Figure 7B*). P170 has expression in the anterior part of the basolateral amygdalar nucelus (BLA, *Figure 7C*) and P113 shows complementary expression in lateral and basomedial amygdalar nuclei (BLA and BMA, *Figure 7D*).

There are 29 lines with expression in striatum and related structures. P189 had expression in the central amygdalar nucleus (CEA, *Figure 7E*) and P161 had expression in the lateral septum (*Figure 7F*). Some lines have regional expression in caudoputamen. For example, P057 has expression only in lateral caudoputamen (*Figure 2H*) and P172 has expression in the striasomes in dorsal caudoputamen projecting to both globus pallidus, external part (GPe) and substantia nigra, reticular part (SNr) (*Figure 7G*). Striatal mCitrine-expressing cells in P118 are biased toward the anterior caudoputamen and appear to consist mainly of indirect pathway cells projecting to GPe (*Figure 7H*). Additional patterns of regional expression in the striatum include lateral (P055, *Figure 2H*), and

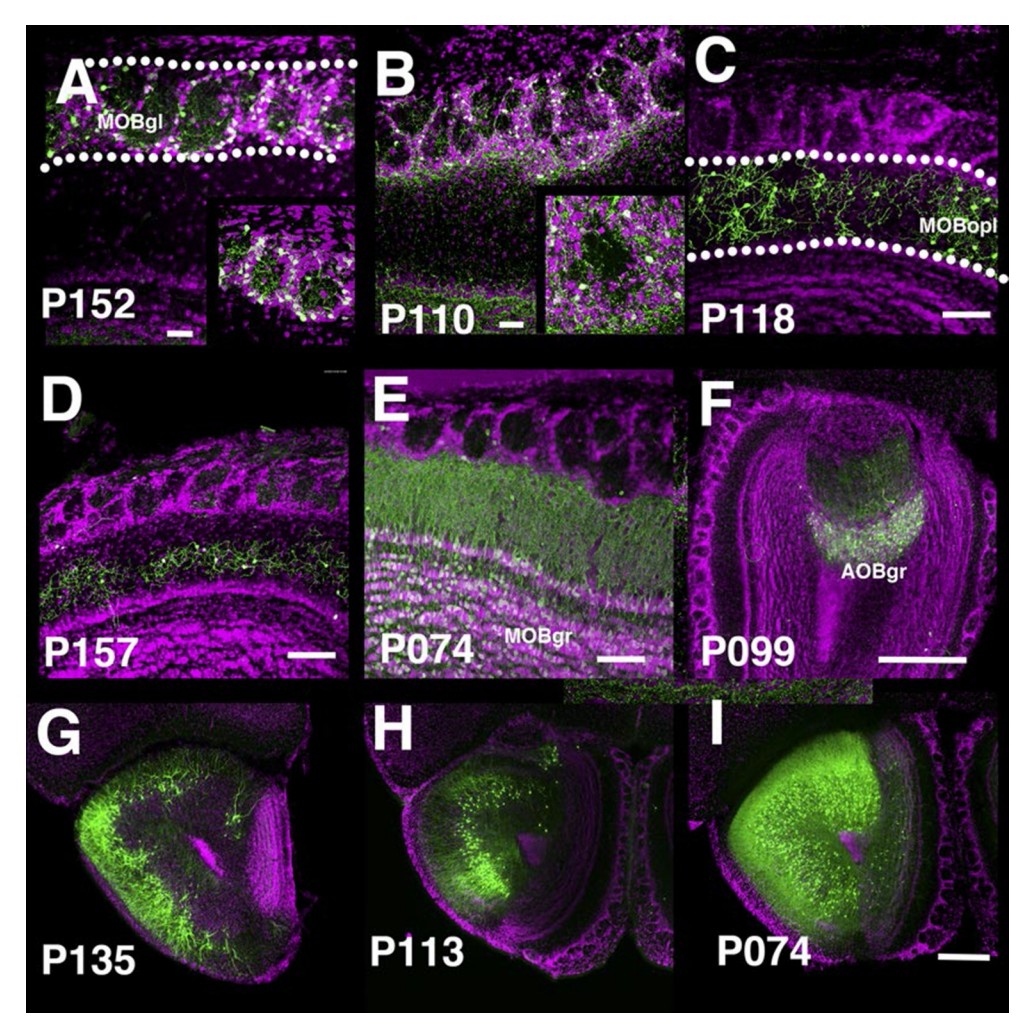

**Figure 5.** PB lines labeling olfactory bulb cell types. Coronal sections of main olfactory bulb (**A–E**), accessory olfactory bulb (**F**), and anterior olfactory nucleus (**G–I**). AOBgr: accessory olfactory bulb, granule layer, MOBgl: main olfactory bulb, glomerular layer, MOBgr: main olfactory bulb, granule layer MOBopl: main olfactory bulb, outer plexiform layer. Scale bars in **A–E**: 100 μm, others: 500 μm.

dorsal striosomes of both direct (projecting to substantia nigra pars reticulata) and indirect (projecting to globus pallidus) pathways (P172, *Figure 7G*), as well as to striasomes restricted mainly to the indirect pathway (P118, *Figure 7H*).

## Thalamus and hypothalamus
We obtained lines with expression in specific thalamic and hypothalamic nuclei, including anterior medial nucleus (P084, *Figure 7I*), ventral lateral geniculate nucleus (P138, *Figure 7J*), submedius nucleus of thalamus (P136, *Figure 7K*) and a part of the paraventricular nucleus of the hypothalamus (PVT, P006, *Figure 7L*). P170 had sexually dimorphic expression; males (Figure 7M1) but not females (Figure 7M2) had expression in the medial preoptic area (MPO) and bed nuclei of stria terminalis (data not shown). Since both structures are larger in males than females (*Cooke et al., 1998*) further work will be needed to determine if the neurons themselves or only their reporter expression are sexually dimorphic. The animals of both sexes had the same expression in other areas.

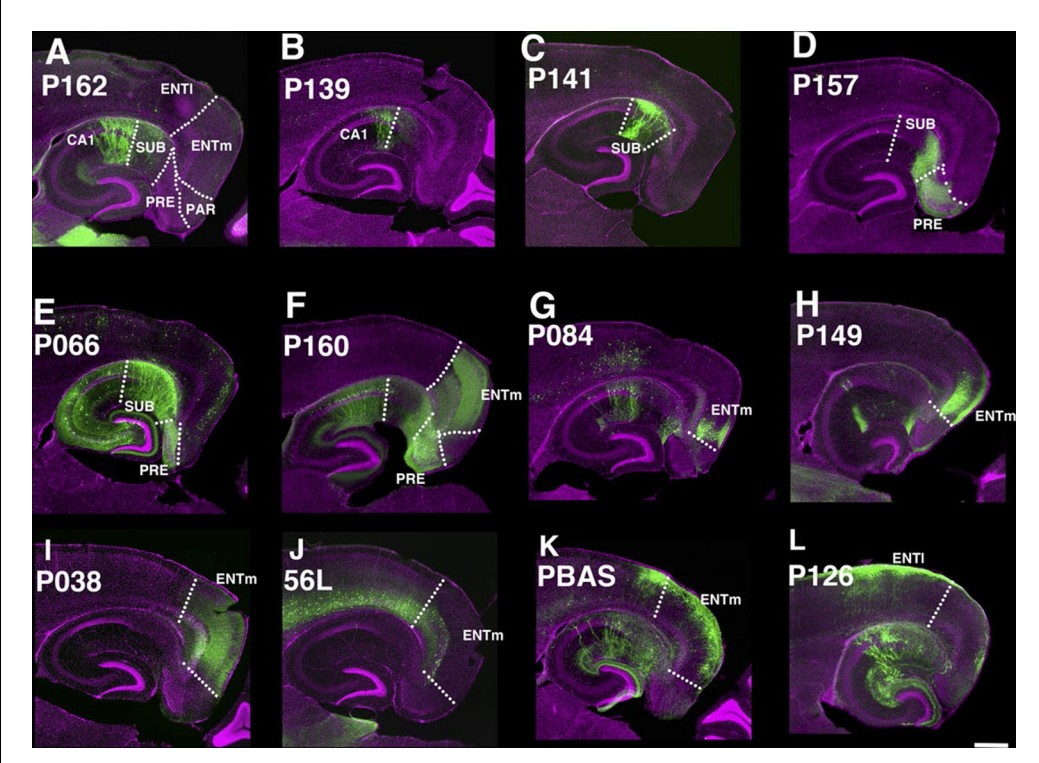

**Figure 6.** Lines with expression in the hippocampal formation. Horizontal sections through the hippocampal formation. (A–B) Expression closer to CA1 (P162, **A**) and to subiculum (P139, **B**) at the region of their border. CA1: Ammon's horn, field CA1, SUB: subiculum, PRE: presubiculum, PAR: parasubiculum, ENTm: entorhinal cortex, medial part, ENTl: entorhinal cortex, lateral part. Arrowheads in (**A–C**) distal end of CA1 pyramidal layer. (**C–E**) Subiculum expression in P141(**C**), P157 (**D**) and P066 (**E**). (**F**) Presubiculum expression in P160. (**G** and **H**) Expression in medial entorhinal cortex layer 5 in P084 (**G**) and P149 (**H**). (**I** and **J**) medial entorhinal cortex layer6 expression in P038 (**I**) and 56L (**J**). (**K** and **L**) medial entorhinal (PBAS, **K**) and lateral entorhinal (P126, **L**).layer 2 expression. Scale bar: 500 μm.
The following figure supplement is available for figure 6:

**Figure supplement 1.** P162 subiculum neurons project to thalamus.

## Midbrain and hindbrain

We observed lines with expression in superior colliculus (SC, PBAU, *Figure 7N*), inferior colliculus (IC, P118, *Figure 7O*). P118 also has expression in the lateral part of the interpeduncular nucleus (IPN, *Figure 7P*) and dorsal medulla, presumably a part of the parasolitary nucleus. In the interpeduncular nucleus, two lines label the dorsal (P025, *Figure 7Q*) and central parts, respectively (P161, see http://enahancertrap.bio.brandeis.edu/P161/coronal/029/5652/4481/25/). P066 has expression in neurons of the inferior olivary complex (IO), which send climbing fibers to the cerebellum (*Figure 7R*). P161 has expression in the solitary nucleus (*Figure 7S*). P118 shows dorsal column nuclei (DCN) expression (*Figure 7T*). P108 has expression in the dorsal part of the spinal nucleus of trigeminal (*Figure 3D2*) and dorsal spinal cord (*Figure 7U*).

## Cerebellum

In the cerebellum, we found many lines with Purkinje cell and granule cell expression. Some lines have broad expression in these cell types (*Figure 8A and D*), and others have sparser expression (*Figure 8B and E*, see also [*Huang et al., 2013*]). Expression in Purkinje cells in P014 was restricted to a small subset occupying a stripe-like pattern (*Figure 8C*) that is consistent across individuals (*Figure 8C* inset). P033 has restricted expression in granule cells that project to the basal half of the

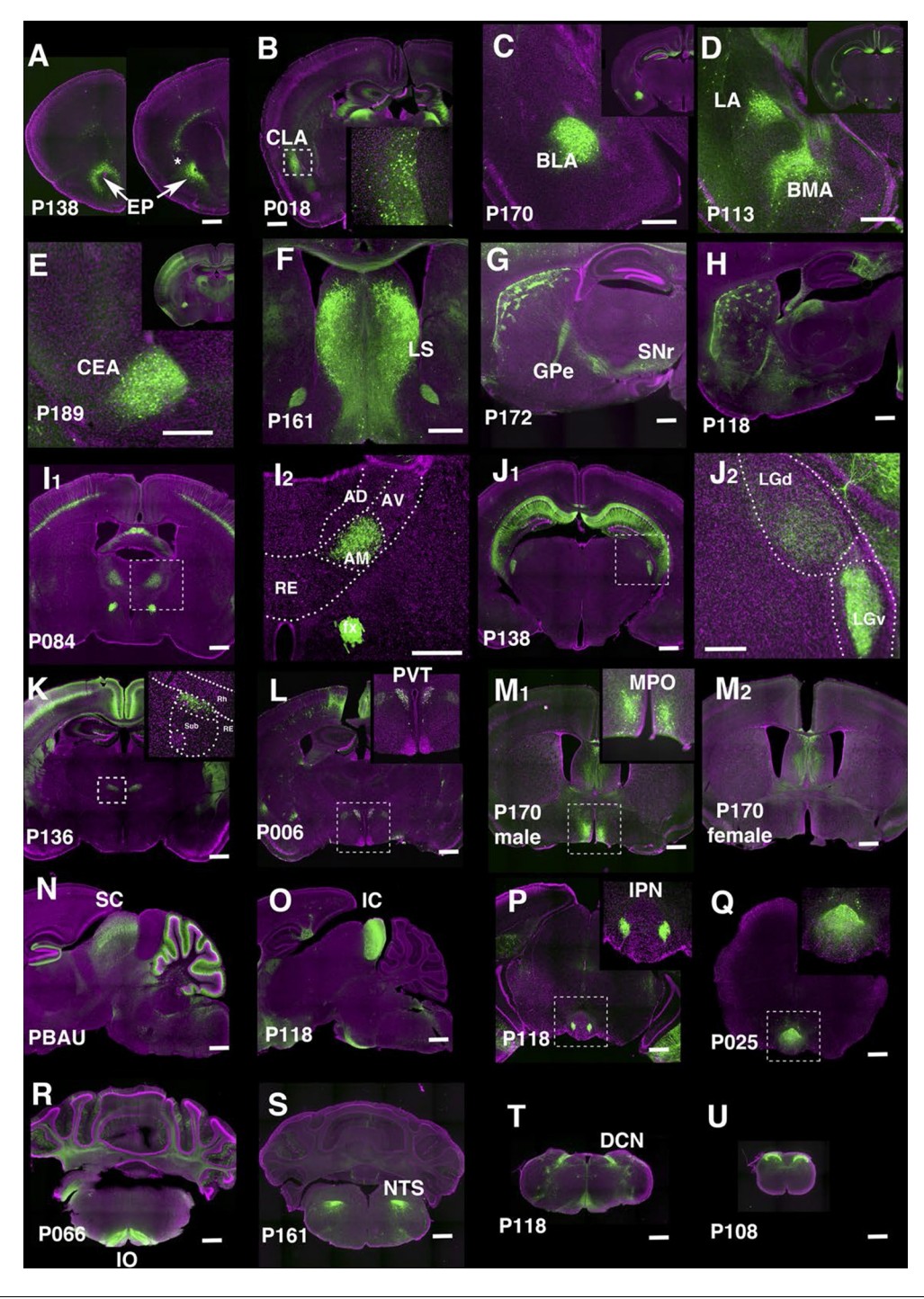

**Figure 7.** Lines labeling cortical subplate, mesencephalic, and diencephalic cell types. (A) Endopiriform nucleus (EP) expression in P138 near anterior olfactory nucleus (left) and claustrum (asterisk). Note claustrum does not express mCitrine. (B) Claustrum (CLA) expression in P018. (C and D) Amygdalar nucleus expression in P170 (C, BLA:basolateral) and P113 (D, LA: lateral and BMA: basomedial). (E) Central amygdalar nucleus (CEA) expression in P189. (F) Expression in P161 lateral septum (LS) (G and H) Striosome expression in P172 (G) and P118 (H). GPe: globus pallidus, external part, SNr: substantia nigra, reticular part. (I) Anterior medial nucleus (AM) expression in P084. I2: close up of the rectangle area in I1. AD: anterior dorsal, AV: anterior ventral, RE: nucleus of reunions (J) Expression in ventral lateral geniculate nucleus in P138. J2: close up of the rectangle area in J1. LGd: dorsal lateral geniculate nucleus. (K) Dorsal submedius nucleus expression in P136. Inset: close-up of the rectangle. Sub:

*Figure 7 continued on next page*

*Figure 7 continued*

submedius nucleus, Rh: rhomboid nucleus. (**L**) Paraventricular nucleus (PVT) expression in P006 (**M**) P170 displayed sexually dimorphic expression (M1:male, M2: female) in medial preoptic area (MPO). (**N**) Superior colliculus (SC) expression in PBAU (**O**) Inferior colliculus (IC) expression in P118. (**P** and **Q**) Expression in subnuclei in interpeduncular nucleus (IPN) in P118 (**P**) and P025 (**Q**) (**R**) Inferior olivary complex (IO) expression in P066. (**S**) Nucleus of solitary tract (NTS) expression in P161. (**T**) Dorsal column nucle (DCN) expression in P118. (**U**) Expression in dorsal spinal cord in P108. Scale bars: 500 µm.

molecular layer. We also found lines with expression in Bergman glia (*Figure 8G*), basket cells (*Figure 8H*), and stellate cells (*Figure 8I*). There were two lines (P134 and P159) that have cell bodies directly beneath the Purkinje cell layer and extend dendrites into the molecular layer (*Figure 8J–L*). From their cell body positions and morphology, they appear to be Lugaro cells.

## Finding a new cell type in piriform cortex

We found three lines (48L, 52L, P113) with distinctive expression patterns in the cell-dense layer (layer II) of the piriform cortex (*Figure 9A–C*). Two broad categories of layer II glutamatergic neurons have previously been described; semilunar (SL) cells, which lack well-defined basal dendrites and are located in the upper sublayer of layer II, and superficial pyramidal (SP) cells, which, like most pyramids, possess distinct basal and apical dendrites and are located deeper in layer II (*Suzuki and Bekkers, 2006*; *2011*). 48L cells were recently shown to be a subset of SL cells (*Choy, 2015*). Based on their cell body positions (*Figure 9D*) and dendritic morphology, the labeled cells in P113 appear to be typical SP cells. 52L cells were GABA-negative (data not shown) but do not clearly match the anatomical and physiological properties of either subtype of previously described pyramidal neurons.

We recorded the physiological properties of mCitrine-labeled cells and non-labeled cells in 52L piriform cortex. Like SP cells, they responded to depolarizing current with an initial high frequency burst of action potentials (*Suzuki and Bekkers, 2006*) as did nearby non-labeled cells (arrows in *Figure 9E* and *8G*). However, labeled cells (but not unlabeled cells) differed from the previously described SP neurons in that they exhibit a stuttering firing pattern and their firing inactivates at higher currents (*Figure 9I*). Labeled and non-labeled neurons also differ in their afterhyperpolarizations (arrowheads in *Figure 9E and G*; *Figure 9J*). While the labeled neurons could not sustain prolonged firing at high-current injections, their instantaneous firing frequency was higher in the beginning of spike train (*Figure 9E and G*; *Figure 9K*). Morphologically, the non-labeled neurons resemble previously described SP cells since they possessed distinct basal and apical dendrites (*Figure 9H*). Labeled neurons also possessed basal dendrites (unlike SL cells) but do not have a distinct apical dendrite (*Figure 9F*). Taken together, our anatomical and physiological results suggest that 52L cells are a distinct subset of SP cells that differ phenotypically from other, unlabeled SP cells.

## An altered classification of layer 6 cortico-thalamic pyramidal neurons

Cerebral cortex and thalamus have dense reciprocal connections and layer 6 of the cortex is the major source of the cortico-thalamic (CT) projection. Single cell tracing has shown that there are two types of L6 CT projecting pyramidal neurons (*Thomson, 2010*; *Zhang and Deschenes, 1997*); primary sensory CT pyramidal neurons send axons to primary sensory thalamic nuclei (ex. ventro-posterior medial:VPm, lateral geniculate dorsal nucleus: LGd). Non-primary CT neurons send weaker projections to primary sensory nuclei, but also project to secondary sensory nuclei; for example, in primary somatosensory cortex, they send axons to VPM, posterior thalamic nuclei (Po), and intralaminar thalamic nucleus. Primary CT neurons are located in upper layer 6 and non-primary CT neurons in lower layer 6. Primary CT neurons also project to the thalamic reticular nucleus (RTN), whereas non-primary CT neurons do not. Primary CT neurons and non-primary CT neurons also have different morphologies. Primary CT neurons extend apical dendrites to layer 4 while dendrites of non-primary CT neurons do not reach to layer 4. Layer 6 also contains corticocortical pyramidal cells, which have long collateral projections within layer 6.

We obtained three lines with expression in layer 6 cortical pyramidal neurons (*Figure 10*). P162 has mCitrine-positive cells in primary somatosensory area (SSp), primary visual area (VISp), and

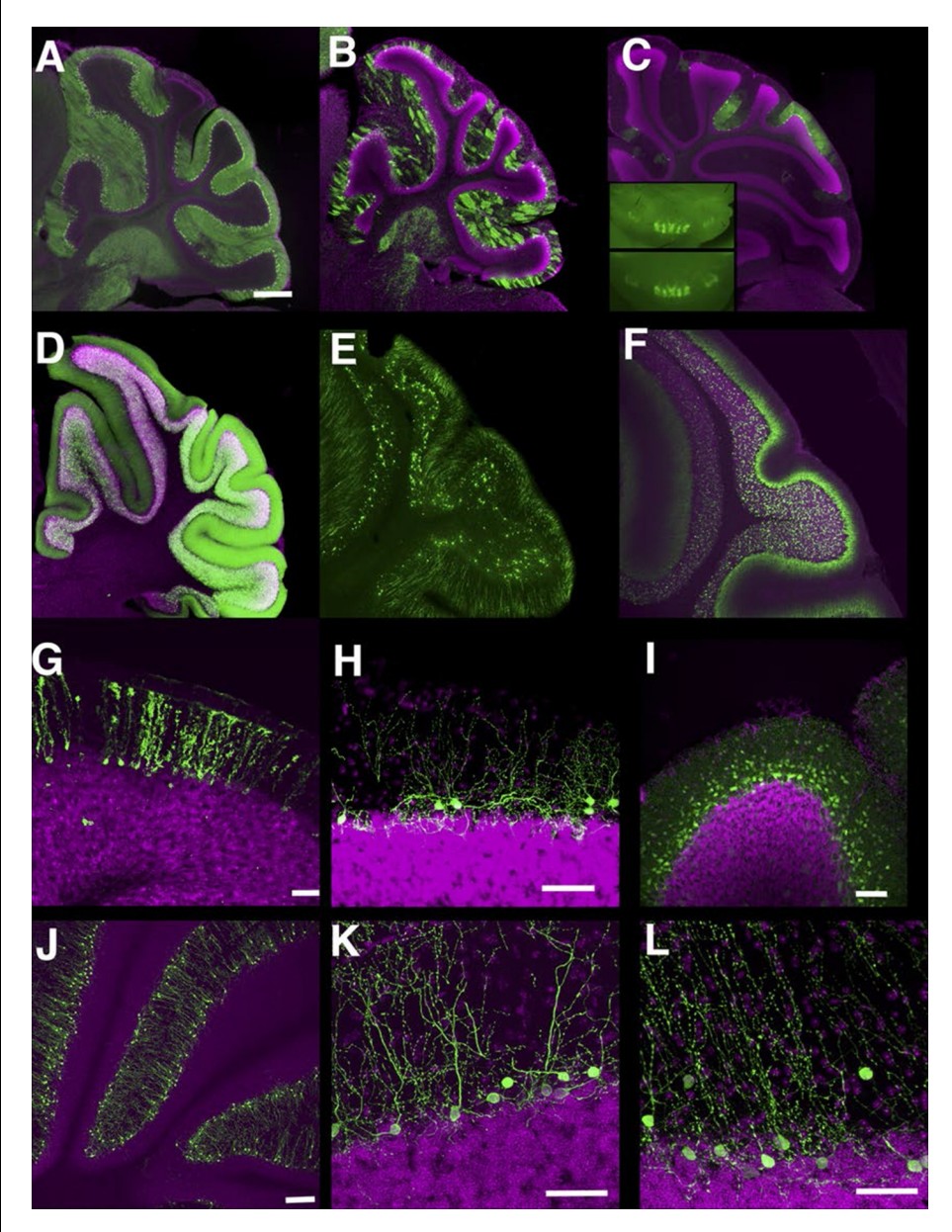

**Figure 8.** Lines labeling cerebellar cell types. (**A–C**) Purkinje cells labeled densely (**A**, P034), sparsely (**B**, P096), and in restricted regions (**C**, P014). **C** inset: dorsal views of cerebellums from two different individuals. (**D–F**) Granule cells labeled densely (**D**, P012), sparsely (**E**, TCGC), and in a population projecting axons to the basal half of the molecular layer (**F**, P033). (**G**) Bergman glia labeling in TCFQ. (**H**) P102 has sparse labeling in basket cells. (**I**) P034 has expression in basket cells and stellate cells. (**J–K**) Lugaro cell like expression in P134 (**J** and **K**) and P159 (**L**). Scale bar in **A–F**: 500 μm, others: 100 μm.

retrosplenical cortex and axonal projection in VPm and LGd (*Figure 10G–J*). P139 has expression in lateral cortex including supplemental somatosensory area (SSs) and gustatory cortex and projection in Po (*Figure 10M–P*). There are topological projection patterns in RTN; dorsally located P162 cells project to dorsal RTN and laterally located P139 cells project to ventral RTN (*Figure 10—figure supplement 1B and C*). The third line, 56L, has broad expression across neocortex (*Figure 10A–D*). 56L neurons have projection to both primary and secondary nuclei but not to RTN (*Figure 10—figure supplement 1A*).

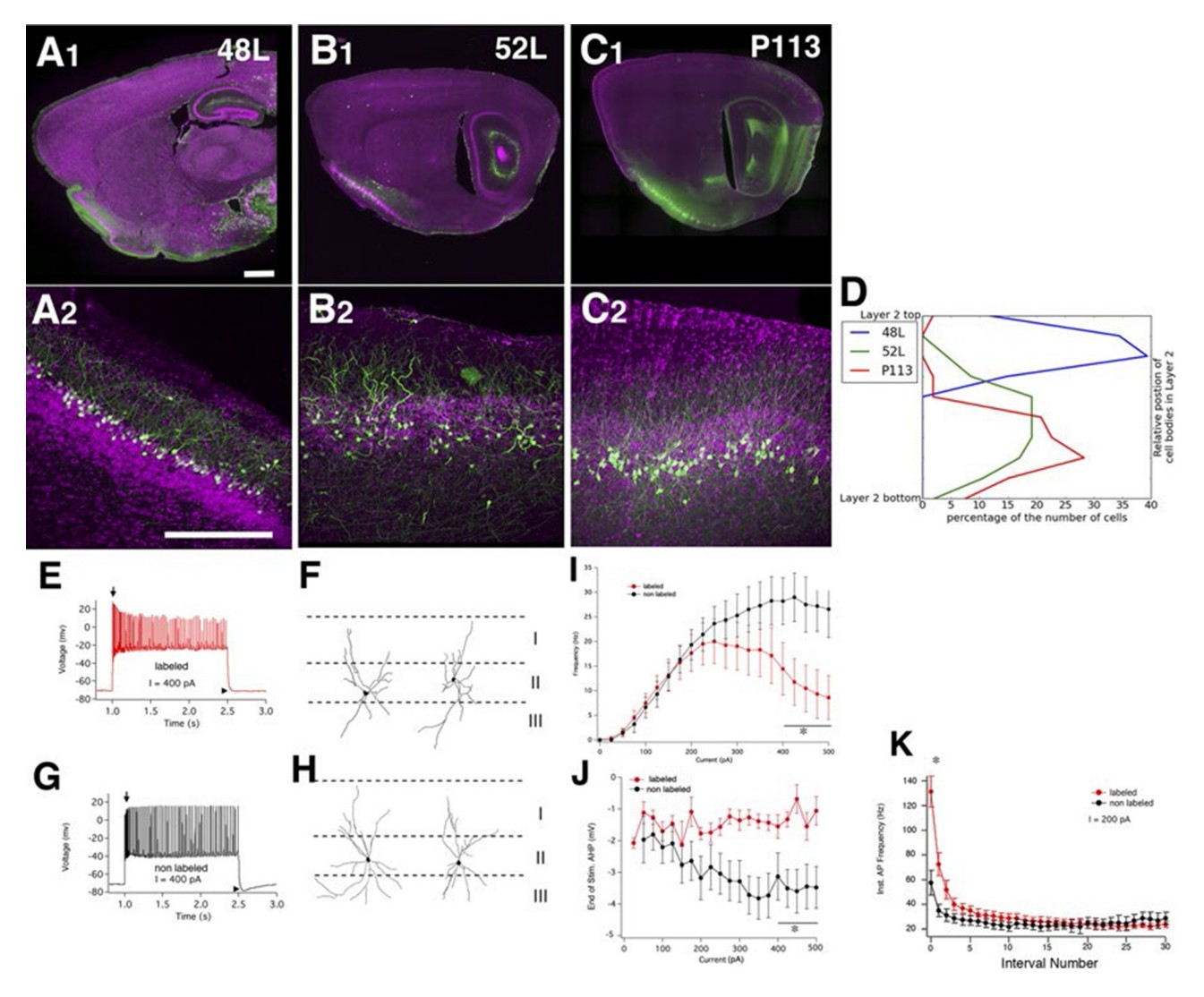

**Figure 9.** Piriform cortex cell types. (A–C) Expression in three distinct populations within piriform cortex. (D) Cell body distributions in layer 2. (E–K) 52L labels a previously undistinguished cell type. Firing patterns (E and G) and morphologies (F and H) of labeled (E and F) and non-labeled (G and H) cells in 52L piriform cortex. Arrows: initial burst present in labeled, but not unlabeled cells' arrowheads: AHP at the end of train present in unlabeled but not labeled cells. Average F–I curves (I), AHP amplitude (J), and instantaneous firing frequency (K) for labeled cells (red) and non-labeled cells (black) were significantly different (asterisks): mean firing frequencies (averaged over 400–500 pA current injection, 11 ± 5 Hz and 28 ± 5 Hz, p = 0.025), AHP amplitude (-1.2 ± 0.3 mV and -3.4 ± 0.6 mV, p=0.0073, labeled and non-labeled cells, respectively), and in instantaneous firing frequencies (131 ± 12 Hz and 58 ± 10 Hz, p = 0.00019). n= 10 for each; line. Scale bars: 500 µm.

The labeled CT neurons also differ morphologically and in their laminar locations. Cell bodies of P162 and P139 are located in upper layer 6, but those of 56L are in lower layer 6 (*Figure 10—figure supplement 2A and B*). P162 and P139 have apical dendrites reaching to layer 4 (*Figure 10K, L and Q*); some apical dendrites even extended to layer 1 (*Figure 10K*). Apical dendrites to layer 1 were more frequently seen in P162 VISp and P139 (*Figure 10L and Q*). On the other hand, 56L dendrites in SSp did not step in to layer 4. In VISp, some neurons extended dendrites to layer 1. From their cell body locations and projection patterns, P162 and P139 appeared to be primary-CT type pyramidal neurons in different cortical areas.

We also recorded the physiological properties of layer 6 cells. 56L cells have larger whole cell capacitances than P162 and P139 cells and tended to have correspondingly lower input resistances. All mCitrine-positive cells fired tonically (*Figure 10—figure supplement 2C and D*). In the 56L

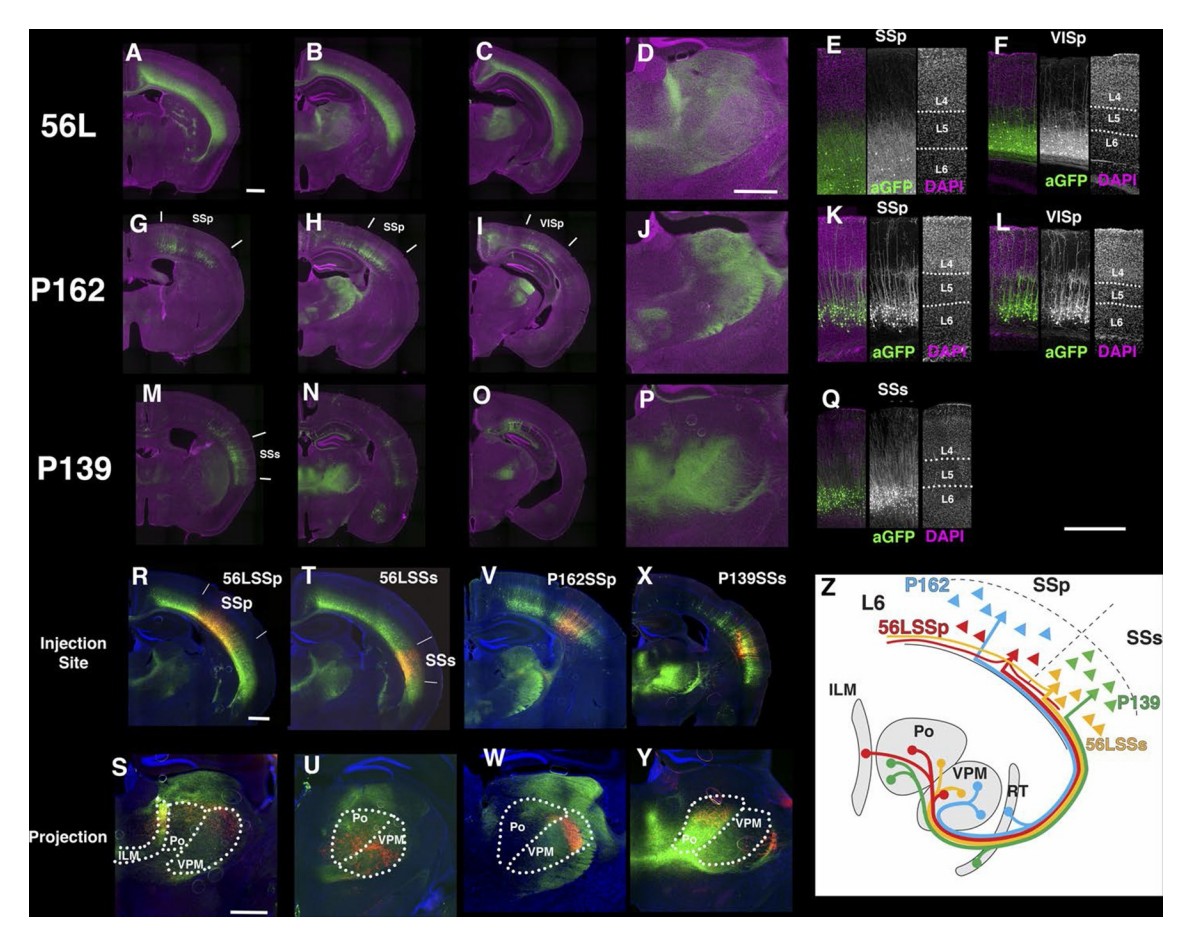

**Figure 10.** Projections of layer 6 corticothalamic (CT) neurons. (A–D) Coronal images from 56L. (E and F) confocal images from SSp (E) and VISp (F) from 56L. (G–J) Coronal sections from P162. (K and L) Confocal images from SSp (K) and VISp (L) from P162. (M–P) Coronal images from P139. (Q) Confocal image from P139 SSs. Sections were taken from 0.7 mm (A, G, and M), 1.7 mm (B, D, H, J, N, and P), 2.3 mm (C, I, and O) caudal from bregma. (R–W) tet-reporter virus injection into 56L SSp (R), 56L SSs (T), P162 SSp (V), and P139 SSs (X) and their projection to thalamus (S, U, W, and Y, respectively). (Z) Schematic view of projections in layer 6 lines. ILM: interlaminar nucleus, Po: posterior complex, VPM: ventral posteomedial nucleus. Scale bars: 500 μm.

The following figure supplements are available for figure 10:

**Figure supplement 1.** Projections to the reticular nucleus of the thalamus (RT) (A–C) DAPI (blue), anti-GFP (green), and anti-Parvalbumin (PV, red) staining for thalamus of 56L (A), P162 (B), and P139 (C).

**Figure supplement 2.** Sublaminar location and intrinsic physiology of layer 6 neurons.

**Figure supplement 3.** 56L axonal projection from VISp to thalamus.

**Figure supplement 4.** Long lateral projections in 56L and P139 AAV–TRE3GmCherryHA was injected to 56L.

recordings, we found that most mCitrine-negative cells recorded near mCitrine-positive cells fired more phasically and at lower rates (*Figure 10—figure supplement 2F–H*) reminiscent of the firing previously described for CC cells (*Mercer et al., 2005*).

We compared axonal projection patterns to thalamus by injecting AAV-TRE3GmyrCherryHA virus in layer 6. P162 SSp cells projected to the dorsal part of VPM and P139 SSs neurons projected to Po (*Figure 10V–Y*). In contrast, labeled axons from 56L SSp cells (*Figure 10R and S*) were found in Po, VPM, and the intralaminar nucleus (ILM), consistent with previously described projections of non-

primary CT cells (*Zhang and Deschenes, 1997*). Axons from 56L SSs were enriched in ventral VPM and Po (*Figure 10T and U*) and those from VISp mainly projected to the lateral posterior nucleus, not to LGd (*Figure 10—figure supplement 3*). We also found that 56L had long lateral axonal projections that even reached to the contralateral hemisphere, whereas P162 and P139 cells had local lateral axonal projections within the cerebral cortex (*Figure 10—figure supplement 4*).

In order to complement our phenotypic analyses of differences between subtypes of L6 corticothalamic neurons, we also analyzed their RNAseq profiles and compared them to VISp layer 6 pyramidal neurons from the Ntsr1-Cre line, which is also known to have layer 6 specific expression in the cortex (*Gong et al., 2007*). Ntsr1-Cre labels virtually all primary CT neurons in layer 6 and also has projection to RTN (*Bortone et al., 2014*; *Kim et al., 2014*; *Olsen et al., 2012*). Clustering samples by correlations between gene expression vectors revealed two main clusters: those from 56L and all others (*Figure 11A*). Samples from P162 and P139 are intermingled in the cluster, implying they have quite similar RNA expression profiles. Analysis of differentially expressed genes also showed clear differences between the two main groups. There were 1869 genes differentially expressed among all sample groups (false discovery rate (FDR) < 0.01), and most differentially expressed genes showed bimodal patterns; high expressions in one group and low expressions in the other (*Figure 11B and C*). We also examined the expression of previously identified layer 6 marker genes (*Molyneaux et al., 2007*; *Zeisel et al., 2015*) and of genes used to generate BAC-Cre lines having layer 6 expression (*Harris et al., 2014*). Most of these known layer 6 markers are expressed both in the Ntsr1-cre group and in 56L (including the Ntsr1 gene itself) or were present only in the Ntsr1-cre lines. None were reliable markers for the 56L population (see *Figure 11—figure supplement 1* and supplemental Note). We also examined expression profile of entorhinal cortical layer 6 cells from P038 in addition to isocortical layer 6 cells. Based on RNAseq expression profiles, P038 cells belonged to the Ntsr1-cre group but expressed unique set of genes (see *Figure 11—figure supplement 2*).

We confirmed the expression of Tle4 (which is expressed strongly in Ntsr1-cre) and Bmp3 (expressed in 56L) by dual-color in situ hybridization. Tle4 and Bmp3 have essentially non-overlapping expression (*Figure 11D–F*). mCitrine-positive cells in P162 and P139 dominantly express Tle4 with only a few Bmp3-positive cells, whereas 56L cells are mostly Bmp3-positive but Tle4-negative (*Figure 11G–O*). In all lines, the majority of marker (Tle4 or Bmp3)-positive cells do not express mCitrine, suggesting the three lines label subsets of these marker-positive cells. We also analyzed Tle4 expression in the Ntsr1-Cre animal (Nstr1-Cre;Ai14 Rosa-TdTomato) from dual-fluorescent in situ images (http://connectivity.brain-map.org/transgenic/experiment/100147520) and found Ntsr1-Cre cells co-expressed Tle4; 100% (333/333) of Tdtomato[+] cells were Tle4[+] and 77.8% (333/428) of Tle4[+] cells were TdTomato[+]. These results support the view that P162 and P139 are subsets of the Ntsr1-cre population and that 56L cells are a distinct population of L6 CT neurons.

## Discussion

We have developed a highly efficient method of enhancer trapping in the mouse and have used it to generate a resource of lines that allow targeting of a wide range of known and novel neuronal cell types. The enhancer trap approach produces more focused labeling than commonly used approaches that attempt to recapitulate known patterns of endogenous gene expression. Using this approach, we have identified dozens of new subtypes of previously identified neuronal cell types and have clarified the classification of pyramidal neurons within the piriform cortex and within layer 6 of neocortex. The approach is readily scalable since new lines can be generated simply by additional rounds of breeding. We also develop the enhancer trap line web browser to search lines of interests and to share images and detailed information about lines. The web site can serve as a useful open resource for wide range of researcher in mouse genetics and neuroscience.

Cell-type-specific patterns of gene expression are thought to reflect interactions between regulatory sequences within the proximal promoter, and at other far more distal sites (*Nathanson et al., 2009*; *Pennacchio et al., 2006*). Viral reporter was mostly expressed only in mCitrine-positive cells, and we did not see major ectopic reporter expression, which supports that cell-type specific tTA expression, but not regional TRE silencing, mainly contributes to highly restricted expression patterns. Single genes often have multiple enhancer modules each of which regulates expression in different regional or developmental contexts (*Dickel et al., 2013*; *Visel et al., 2009*). By harnessing

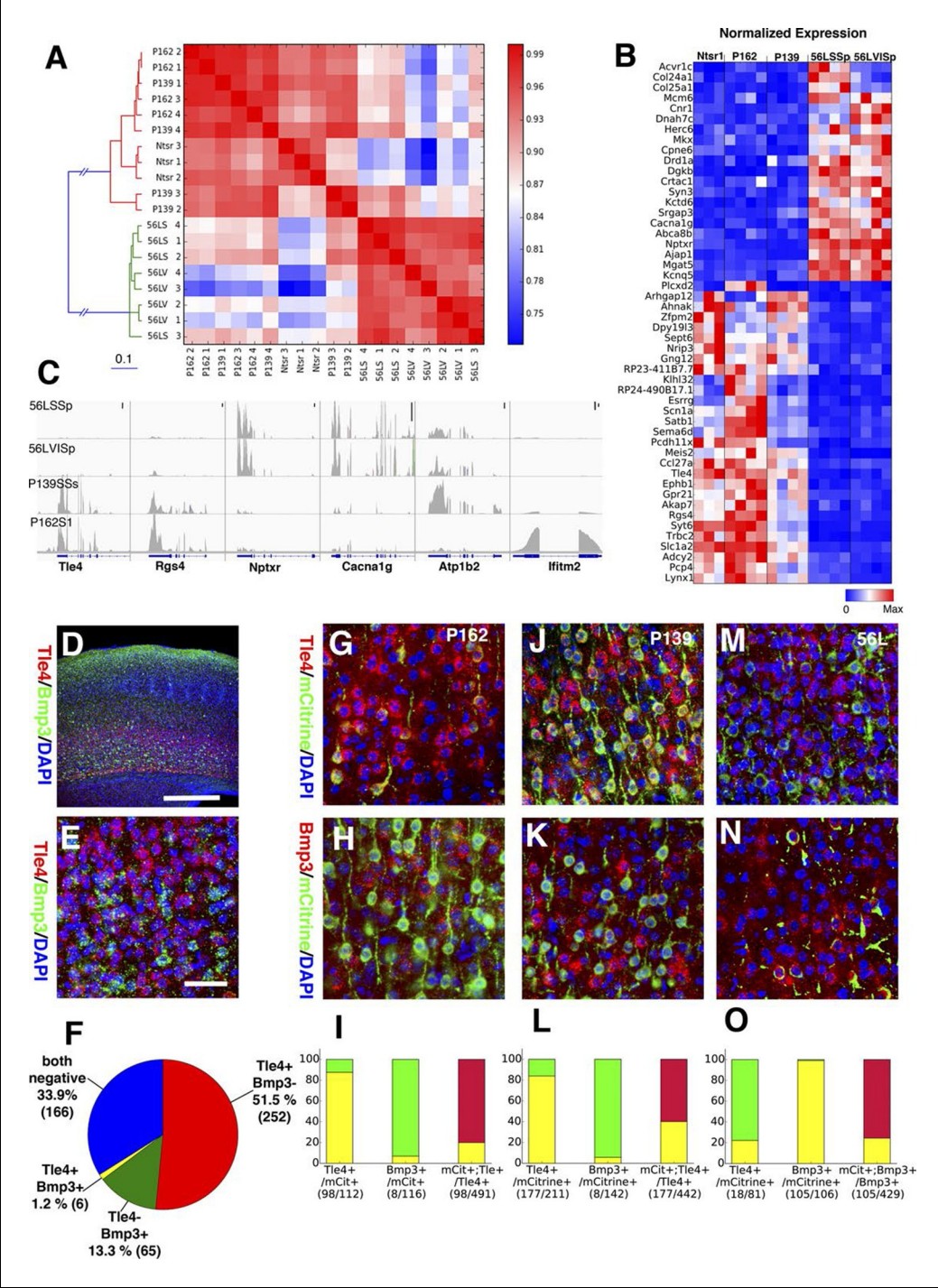

**Figure 11.** Two main subtypes of L6 CT neurons distinguished by gene expression . (A) Clustering of L6 CT neuron samples based on correlations (color scale) between expression profiles. (B) Heat map of normalized gene expression (TPM) of 50 genes with lowest ANOVA p-values. Except for Plcxd2 (asterisk), the genes had dominant expression in either Ntsr1/P162/P139 or 56L. (C) Coverage histograms of differentially expressed genes. Examples of genes expressed in P162/P139 (Tle4 and Rgs4), 56L (Nptxr and Cacna1g), P139 (Atp1b2), and P162 (Ifitm2). Scale bars: 100 counts. (D–F) In situ hybridization for Tle4 (red) and Bmp3 (green) in wild type P10 animal SSp. (E) high-magnification image. (F) Proportion of cells expressing Tle4 and Bmp3 in SSp layer 6. (G–O) In situ hybridization for mCitrine and Tle4 (G, J, and M) or Bmp3 (H, K and N) in P162 SSp (G and H), P139 SSs (J and K) and P56 SSp (M and N). (I, L, O) Proportions of mCitrine$^+$ cells that expressTle4 or Bmp3 and converse proportions of cells expressing the dominant marker (Tle4 for I,L Bmp3 for O) that are mCitrine$^+$ from P162 (I), P139 (L) and 56L

*Figure 11 continued on next page*

*Figure 11 continued*

(**O**). Colors in bar graphs represent in situ signal patterns (Red: cells with marker gene but not mCitrine, Green: cells with mCitrine signal but not marker gene, and Yellow: cells with both marker and mCitrine signals). Scale bar in D: 500 µm, in E: 50 µm.

The following figure supplements are available for figure 11:

**Figure supplement 1.** Expression of known L6 marker genes.

**Figure supplement 2.** P038 entorhinal cortex layer 6 neurons are a distinct population.

these distal enhancers, BAC-transgenic (*Gong et al., 2003*; *2007*) and knock-in (*Taniguchi et al., 2011*) approaches have been used to generate lines that copy expression patterns of targeted genes with high-fidelity, but many of these are quite broadly expressed (*Harris et al., 2014*; *Madisen et al., 2015*). On the other hand, lines with more limited expression patterns have been generated by unintentional positional effects arising from transgene insertion sites. Restricted expression patterns in the series of Thy1 lines (*Feng et al., 2000*) and CA1-specific Cre mouse with a CamKII-promoter (*Tsien et al., 1996*) are notable examples. Because of these positional effects, even BAC transgenic lines occasionally have more restricted expression patterns that differ from those of the targeted genes (*Huang and Zeng, 2013*). Although positional effects can restrict expression to specific populations, the population targeted is not predictable because the enhancer code that directs expression to specific cell types is not well understood. To circumvent this limitation, a useful enhancer trap requires screening a large number of individual strains. This has previously been done using transposable element mobilization in flies and fish (*Balciunas et al., 2004*; *Bellen et al., 1989*; *Brand and Perrimon, 1993*; *Scott et al., 2007*; *Urasaki et al., 2008*). However, since pronuclear-injection produces a rather low efficiency of transgenesis, this approach has not frequently been used in mouse genetics.

Hopping from single copy PB enabled fast identification of insertion sites without the laborious and time-consuming steps of tracking and segregating multiple transgene alleles. In fact, since we have made both single copy PB insertion lines and a line carrying PBase in the male germ line freely available, other laboratories can now screen for additional lines of interest without needing to isolate or inject embryos. Our strategy of including a mCitrine reporter on the probe enables fast expression screening without crossing to a separate reporter line, and including tTA enables inducible genetic manipulation in specifically labeled populations.

Although our enhancer trap lines were generated by random insertion, most lines maintain consistent expression over generations. In addition, lines generated by local hopping have similar expression patterns. These facts support the conclusion that the expression patterns are not generated randomly, but instead are tightly linked to transgenes' insertion sites. PB translocation sites were widely distributed over the genome and lines with insertions far away from known genes often exhibited specific expression presumably by trapping distal enhancers. Distal enhancers are known to regulate tissue specific expression of genes, especially in the forebrain (*Nord et al., 2013*; *2015*; *Visel et al., 2009*; *2013*) and to regulate activity-dependent gene expression (*Kim et al., 2010*). Our results suggest that enhancers are also involved in the fine-grained specification of cell types, and that trapping them can cause very restricted expression patterns. Indeed, drivers restricted to major cell types and layers in laminated structures are already available, but drivers that pick out cell types in specific cortical regions, or thalamic nuclei are quite rare with gene-based strategies but were more common with our enhancer trap strategy.

The cell types accessed genetically in these and other driver strains are best thought of as operational cell types defined by the intersection of a driver strain and an anatomical region. This permits reproducibility but does not define the full set of 'atomic' cell types that comprise the nervous system. Like brain regions, cell types may be arranged hierarchically into tree-like taxonomies. Most operational cell types represent branches or nodes that can be further subdivided. This subdivision occurs when properties such as morphology, physiology, projections and gene expression are found to vary discontinuously. Eventually, those that cannot be further subdivided may be thought as terminal branches or 'atomic cell types'. For a few of the cell types identified in our enhancer trap

strains (e.g. subsets of corticothalamic neurons or piriform cortex neurons) further characterization demonstrates functional distinctions between closely related subtypes. In other cases, the trapped neurons correspond to well-characterized subtypes of a larger class (e.g. LGN thalamic projection neurons), while in many other cases additional characterization will be needed to determine how trapped subtypes differ functionally from other cells in the same class. The anatomical distinctiveness of, for example, Purkinje cells restricted to particular folia or granule cells sending their axons to particular sublaminae are suggestive, but whether these neurons differ from other Purkinje cells and granule cells in other aspects of their anatomy, physiology and gene expression remains to be seen. Efforts to enhance the aggregation of such phenotypic data are needed to better refine the definition of cell types within the vertebrate nervous system. Hopefully, additional iterations of our enhancer trap database will benefit from enhanced informatic efforts to improve usability and interoperability with other databases and to make it easier for the community to contribute data that will help parse operational cell types into atomic cell types.

In addition to positional effects, the nature of our enhancer probe might have contributed to producing restricted expression patterns. We obtained lines with specific expression more frequently than those of previous enhancer screening with minimal HSP promoter-lacZ and thy1 promoter-Cre lines, although they employed the same random insertion method (*Kelsch et al., 2012*). Some PB lines with intronic insertions expressed the reporter in neurons in which the inserted genes were strongly expressed. From these facts, we speculate our enhancer probes may require a certain level of transcriptional activity to drive the reporter expression, and this thresholding effect may limit the expression of the reporter. The use of a transactivator system, rather than a recombinase system, may also have contributed to generating specific reporter expression. Cre-dependent reporter expression will be present regardless of whether cre continues to be expressed, or whether it was only expressed in the labeled cells or their progenitors earlier in development (*Harris et al., 2014*), while tet reporters will be expressed only when tTA protein continues to be expressed.

Our screen revealed two major subtypes of corticothalamic pyramidal neurons in layer 6: primary CT neurons and non-primary CT neurons. The two cell types differ in distribution within layer 6 and have distinctive axonal projection patterns in the thalamus. They also have distinct RNA expression profiles identifying marker genes that display almost non-overlapping patterns in layer 6. Genetic labeling of primary CT cells by Ntsrt1-Cre (*Gong et al., 2007*) has greatly advanced understandings of functions of layer 6 primary CT in cortical (*Bortone et al., 2014*; *Kim et al., 2014*; *Olsen et al., 2012*) and thalamic (*Crandall et al., 2015*) circuits. Perhaps since Ntsr1-Cre labels nearly all (92.7% in SSp and 95–100% in VISp :*Bortone et al., 2014*; *Kim et al., 2014*) primary thalamic nuclei projecting CT neurons, and since a selective driver for non-primary CT layer 6 neurons was not previously available, the role of this second CT pathway from L6 has not been taken into consideration in previous functional studies (*Bortone et al., 2014*; *Kim et al., 2014*; *Watakabe et al., 2014*; *Yamawaki and Shepherd, 2015*). We found that 56L cells in SSp had projection to multiple nuclei (VPM, Po and ILM) as originally described by Zhang and Deschenes (*Zhang and Deschenes, 1997*). 56L in SSs has strong projections to VPM, which implies there are previously unidentified sources of CT feedback from higher order cortical regions to primary sensory thalamic nuclei (*Sherman, 2012*; *Thomson, 2010*). We also found that 56L cells have long collateral projections like those previously described for CC neurons (*Zhang and Deschenes, 1997*). Thomson and colleagues found that the most of cells with CC-like morphologies fired phasically (*Mercer et al., 2005*). We speculate that 56L-like cells are the minor population of layer 6 pyramidal neurons which fire tonically like primary CT neurons but which possess large collateral projections and other morphological features associated with CC cells. Since mCitrine-negative cells in 56L were phasically firing, we speculate that these are CC cells labeled by neither Ntsr1-Cre nor 56L. For example, many of the Bmp3-positive cells not labeled by 56L could be CC-cells lacking a projection to thalamus.

Although we have shown the ability of the tet enhancer trap system to label highly restricted specific cell types, further technical improvements may enhance the utility of the approach. Replacing the HSP promoter with other (minimal) promoters may change the forebrain bias (see **PB line expression patterns** above) to permit better exploration of cell types in other major brain regions. Developments of additional molecular genetic tools, such as optogenetic tools (*Choy et al., 2015* is an example), voltage- or calcium sensors, and viral vectors targeted to synapses or other subcellular structures,or functionalized for retrograde or transsynaptic transport may enhance analysis of the connectivity of trapped cell types. Finally, enhancer trap lines may be useful for analyzing the function of

candidate enhancers near the insertion sites in order to better understand how distal enhancers contribute to the specification and maintenance of cell-type-specific gene expression in the mammalian nervous system.

## Materials and methods

### Lentiviral transgenesis

Lentiral constructs were made using a backbone from pSico (Addgene, Cambridge, MA #11578). Lentiviruses were prepared and injected into single cell embryos as described previously (*Lois et al., 2002*) using virus solutions at $10^9$ infection unit/ml. Candidate forebrain enhancer sequences were chosen using the VISTA enhancer browser (http://enhancer.lbl.gov). The four selected sequences (hs119, hs121, hs122, and hs170) were amplified from C57Bl6/J genomic DNA.

### PiggyBac transgenesis

pPB-UbC.eGFP (*Yusa et al., 2009*) was used as the backbone for PB plasmids. Prior work has distinguished two functional types of insulators: 'barrier' insulators, which prevent the spread of DNA methylation and silencing, and 'blocking' insulators, which limit promoter-enhancer interactions (*Gaszner and Felsenfeld, 2006*). Most vertebrate insulators with barrier activity also have blocking activity (*West et al., 2002*). The cHS4 site from the chicken β-globin locus is a well-characterized insulator known to have separate sequences that mediate its blocking and barrier insulator functions (*Dickson et al., 2010*). To prevent silencing but not enhancer effects, we synthesized cHS4 sequence without the region responsible for the blocking activity (the CTCF-binding site). Tandem copies of the insulator sequences were inserted into each 5' and 3' ends of PB constructs (HS4 ins, *Figure 1—figure supplement 2B*).

The Rosa-PBase line was provided by Ronald Rad and Allan Bradley (*Rad et al., 2010*). In order to establish the Protamine1 promoter-hyPBase line, the 848 bp mouse Protamine1 promoter (*Zambrowicz et al., 1993*) was amplified by PCR from C57Bl6/J genomic DNA and fused with a hyperactive PiggyBac transposase (*Yusa et al., 2011*) and the SV40 polyadenylation signal. The linearized DNA was injected to pronuclei of single-cell embryos. PiggyBac seed lines were generated by pronuclear or cytosolic injection of a PiggyBac plasmid (2 ng/μl) and hyPBase mRNA (50 ng/μl) that was synthesized with mMESSAGE mMACHINE T7 Ultra Kit (Life Technologies) and purified with MEGAClear (Life Technologies).

### Amplification of insertion sites by ligation mediated (LM)-PCR

was performed as described by Wu et al. (*Wu et al., 2003*). We used the same adaptors and primers to amplify lentiviral and PB insertion sites. In addition to the adaptors, the following primers were used for PiggyBac lines; PB5'LMPCR: 5'-CGGATTCGCGCTATTTAGAA-3', PB5'LMPCRnested: 5'-TCAAGAATGCATGCGTCAAT-3', PB3'LMPCR: 5'-CCGATAAAACACATGCGTCA-3', PB3'LMPCRnested: 5'-CGTCAATTTTACGCATGATTATCT-3'. After nested PCR, amplified products were isolated by agarose gel electrophoresis, and then reamplified by PCR to remove non-specific products. The final PCR products were used as templates for direct sequencing with the nested primers. Insertion sites were mapped on C57Bl6/j genome (GRCm38/mm10) with blat (http://genome.ucsc.edu/cgi-bin/hgBlat).

### AAV preparation

For nanobody-split Cre construction, GBP1 and GBP6 (Addgene #50791 and #50796, *Tang et al., 2013*) were fused with NCre and CCre from split Cre (gifts from *Hirrlinger et al., 2009*). AAV purification was performed as described previously (*Zolotukhin et al., 1999*). Since AAV serotypes can show tropism for specific cell types, we used a cocktail of 4 serotypes (2/1, 2/5, 2/8, 2/9). After iodixanol step gradient, the virus solution was dialyzed and concentrated with Amicon Ultra 100k Da filters (EMD Millipore, Billerica, MA) with lactated Ringer. Virus copy number was quantified with real-time PCR. Virus titers were in the range of $10^{12-14}$ gene counts/ml.

## Stereotaxic injection

We followed surgical procedures previously described in (*Cetin et al., 2006*). For each injection, 30-50 nl virus solutions were injected to the target sites with a custom-made injector.

## Physiology

Whole cell recordings from visually identified neurons were obtained as previously described (*Miller et al., 2008*). We recorded from four or more animals for each condition. We used t-test for statistical analyses if not stated.

## Histology

After being deeply anesthetized with Ketamine and Xylazine, mice were perfused with phosphate buffer saline (PBS) and 4% paraformaldehyde in PBS. Brains were post-fixed overnight with 4%A PFA/PBS, embedded in 2% agarose /PBS, and then sectioned at 50 μm with a vibratome (Leica, Buffalo Grove, IL VT1000S). The following antibodies were used for immunohistochemistry: anti-GFP rabbit (Life Technologies, Thermo Fisher, Waltham, MA A-11122), anti-GFP chicken (Aves labs, Tigard, OR GFP-1020), anti-HA rat (Roche diagnostics, Indianapolis, IN clone 3F10). Whole slide images were taken with a microscope with 5x objective and XY-stage controlled by μManger (https://micro-manager.org). Grid/Collection stitching Fiji plugin (*Preibisch et al., 2009*) was used for image assembly. We followed the dual-color in situ protocol described in BraInSitu web site (http://www.nibb.ac.jp/brish/indexE.html) (*Watakabe et al., 2006*).

## Evaluation of expression

The anatomy structure model in Allen reference mouse atlas (http://mouse.brain-map.org/static/atlas) was used to annotate expression areas. To compare the expression areas with Cre lines, 228 structures used in annotation commonly with sagittal and coronal sections in *Harris et al., 2014* were applied. All lines were annotated by three observers independently and the unions of annotations were used. Expression levels were determined by the level of localization in anatomical structures and density of mCitrine-positive cells (*Figure 3—figure supplement 1*). Cell densities were determined by counting cells in most zoomed images in the web viewer (the window size is 600 x 400 px, 768 x 500 μm with images taken with a x5 objective). Structures with more than 10 cells/$mm^2$ (4 or more cells in the window) were annotated.

## RNA-seq

Manual sorting of fluorescent-labeled cells from transgenic animals was performed as described previously (*Sugino et al., 2006*). Total RNA was extracted from manually sorted cells (<200) with Picopure RNA isolation kit (Thermo Fisher), and RNA-seq libraries were made with Ovation RNASeq System V2 and Encore kit (NuGEN, San Carlos, CA). Three or four biological duplicates were made for each sample. Illumina (San Diego, CA) HiSeq2500 was used for sequencing. rna-STAR (*Dobin et al., 2012*) and cufflinks 2.1 (*Trapnell et al., 2010*) were used for mapping reads to reference mouse genome GRCm38 and for transcriptome assembly and quantification, respectively. Gene counts data generated with HT-seq {*Anders, 2015* #103} was used for differentially expressed gene analysis by edgeR (*Zhou et al., 2014*). Custom-written Python programs using numpy and scipy were used for analysis. The accession number of RNAseq data is GSE75229.

## Web page

Following programs were used to build the web site; Python 3.4 (programming language), Django 1.8 (web application framework), mySQL 5.6 (relational database), haystack-2.3.1 (search module for Django), elasticsearch-1.4.4 (search engine), uwsgi-2.0.10 (WSGI application server), and nginx-1.8 (web server).

## Supplemental note

### Lentiviral constructs

The tet transactivator (tTA) was driven by the minimal promoter from the mouse heat shock protein 1A (Hspa1a) gene (*Bevilacqua et al., 1995*), which has previously been used for enhancer trapping in fish (*Bayer and Campos-Ortega, 1992*) and mice (*Kelsch et al., 2012*). Among the constructs

tested, the hsp construct (hsp-tet) was most efficient at generating lines with brain expression (28.8%, see *Table 1*) and many had restricted expression patterns (See *Supplementary file 1* for detail). We also tested hsp-probes with a spacer between tTA and TRE (hsp-tet2) and with inverted orientation (hsp-tet3). All hsp-tet2 animals had either broad or no brain expression and hsp-tet3 did not generate any lines with brain expression.

We also attempted to bias expression patterns by including candidate enhancers from highly-conserved sequences (*Pennacchio et al., 2006*) or by using promoters of genes known to have more restricted expression patterns (*Figure 1—figure supplement 2A*). Specifically, we tested four ultra conserved, non-coding sequences found to have forebrain enhancer activity in an embryonic screen (*Visel et al., 2007*) and four promoters—CamKII (*Tsien et al., 1996*), Gad1 (*Chattopadhyaya et al., 2004*; *Szabo et al., 1996*), Thy1 (*Feng et al., 2000*), and Slc32a (*Ebihara et al., 2003*) — previously used to establish cell-type specific transgenic lines. Most of these enhancer and promoter lentiviral probes had lower rates of brain expression rate than the hsp-tet probe (*Table 1*). The CamKII promoter probe had comparable efficiency to hsp-tet but most lines had broad expression in the cerebral cortex (data not shown).

## PB Translocation

Previous studies have shown that piggyBac preferentially integrates into genes and other regions of active chromatin, and has a much weaker tendency hop locally than SleepingBeauty (*Li et al., 2013*; *Liang et al., 2009*). We analyzed the patterns of translocation (*Table 3*; see *Supplementary file 1* for insertion sites of each line). The proportion of insertions into genes (60/167, 35.9% ) was comparable to that expected by chance (33.3% (*Liang et al., 2009*). We expected that prm-PBase may allow integration into a wider range of target locations than Rosa-PB because PB is expressed during histone-to-protamine transition that occurs in spermatogenesis. But Rosa-PBase (15:29) and prm-PBase (45:78) had similar gene-intergenic translocation ratios. Except for local hops, there were no particular genomic regions enriched for insertions and there were no insertions into previously described hot spots for PB translocation in mouse ES cells (*Li et al., 2013*). The frequencies of local hopping and reinsertion within the same chromosome (*Table 4*) were comparable to the rates previously observed with SleepingBeauty (*Ruf et al., 2011*).

We found that Prm-PBase generated founders more efficiently than Rosa-PBase (*Table 2*). Prm-PBase conferred higher transposition rates than Rosa-PBase. In addition, the restriction of PBase to the male germ line meant that more founders could be established from Prm-PBase than from Rosa-PBase. Females of PB;PrmPBase did not have PBase activity and therefore could be founders of new lines whereas both sexes of PB;Rosa-PBase would not be able to transmit PB alleles stably (see differences in efficiency of new line production rate in *Table 2*). We also found that all PB + founders from PBAW/Y;Prm-PBase/+ were females even though PB jumped to many other chromosomes. This showed tight regulation of Prm1 promoter only in meiosis II where sex chromosomes were already segregated. P2 animals with Prm-PBase allele and those without the allele had similar PB transposition efficiency, implying PB translocation occurred in sperm that did not carry the Prm-PBase gene. As suggested for Prm-SleepingBeuty (*Ruf et al., 2011*), PBase proteins may be supplied from Prm-PBase-positive sperms through cytoplasmic bridges among spermatids.

## PB insertion sites and reporter expression patterns

In order to determine if the reporter expression patterns reflected those of genes near the insertion sites, we examined the expression patterns of nearby genes using available databases (Allen Brain Atlas: http://mouse.brain-map.org and gene expression database: http://www.informatics.jax.org/expression.shtml). In most cases, in situ signals were broad and/or weak and were not strongly correlated with reporter expression. We often found clear correlation of expression patterns in the lines with intronic insertions. P103, for example, had strong reporter expression in hippocampus and local expression in a subset of Purkinje cells, whereas the inserted gene Gria1 has enriched expression in the hippocampus and nearly all Purkinje cells in the cerebellum. P062 has insertion in Slc9a2 intron and both P062 and Slc9a2 have strong expression in CA1.

## Viral reporter expression and 'leak' from tet promoters

We devoted significant effort to developing reagents that would allow us to 'convert' tet to cre expression in our driver strains. We first checked the expression of tet reporters in 293T cells. We found that TREtight-myrmCherry (myristoylated mCherry) had strong tTA-independent (leak) expression (compare *Figure 2—figure supplement 3A1 and A2*). The leak expression from TRE3G (a third generation TRE with reduced leak expression) was hardly detectable (*Figure 2— figure supplement 4A and B*). We also evaluated leak in vivo. We injected AAV constructs containing the tTA reporter into the brains of our tet lines and did not observed leak expression (i.e. there was no reporter expression in mCitrine negative cells). Reporter driven by a second-generation TRE (TREtight-myrmCherry-HA) was expressed only in mCitrine positive cells and could be used to map their axonal projections (ex, *Figure 11*). Presumably, the lower leak in vivo reflected a lower copy number of the TRE construct than achieved in cell culture. However, attempts to use TRE to drive recombinase expression revealed leak expression not visible with the tet reporter. Leak expression from TREtight-Cre and TRE3G-Cre were both strong enough to drive the Cre reporter, FLEX-mCherry in mCitrine negative cells both in culture (*Figure 2—figure supplement 4E and F*) and in vivo, where co-injection of TREtight-Cre and Cre reporter (AAV-Flex-mCherry) had non-specific expression in the brain.

We next tried a split-Cre construct, in which the Cre coding sequence is divided into N-terminus (NCre) and C-terminus (CCre) parts and their dimerization via a leucine zipper re-forms the functional enzyme (*Hirrlinger et al., 2009*). Tre3G split Cre AAV (TRE3G-NCre and TRE3G-CCre) did not have detectable leak expression in 293T cells (*Figure 2—figure supplement 4G*) or in the brain (*Figure 2—figure supplement 5C*). We also made a TRE3G flippase (TRE3G-Flpe) and and an FRT reporter construct (AAV-fDIO-myrmCherryHA). The Flippase had no leak expression in cultured cells (*Figure 2—figure supplement 4H*). In the brain, reporter was primarily expressed in mCitrine-positive neurons; however, there were a few mCitrine-negative but FRT-reporter-positive cells (*Figure 2—figure supplement 5D*) indicating low-level leak.

We tested whether the TRE3G-split Cre could drive expression of a floxed reporter. We found that TRE3G splitCre had strong non-specific reporter expression in P113; Ai14 (TdTomato Cre reporter allele) animals (*Figure 2—figure supplement 5D*) indicating that this construct also had an unacceptable level of leak when used with a chromosomal reporter, even though it appeared not to leak when used with a viral reporter.

Finally, we applied GFP-nanobodies to split-Cre (*Tang et al., 2013*). We made fusion protein constructs with GFP-binding proteins (GBP) and NCre or CCre so that functional Cre enzymes would be formed on GFP proteins. Co-expression of GFP-nanobody NCre (TRE3G-GBP1-NCre) and CCre (TRE3G-GBP6-CCre) had tTA independent reporter recombination in cultured cells (*Figure 2— figure supplement 4I*), probably due to GFP-independent alpha-complementation of Cre protein. However, Tre3G-GBPsplit-Cre had specific expression in P113; Ai14 brain (*Figure 2— figure supplement 5F*). In addition, co-injection of TRE3G-GBP-splitCre AAV with Cre reporter AAV had restricted expression only in mCitrine+ cells in 56L (*Figure 2—figure supplement 5G*). Hence using split cre with GFP nanobodies can produce specific cre reporter expression without leak in vivo using both viral and chromosomal reporters.

## Acknowledgements

The transgenic lines were generated at the Brandies University transgenic mouse facility. We are grateful to Frank Sangiorgi and Zhe Meng for transgenesis; Serena David and Rajani Shelke for virus production; Roman Pavlyuk, Hao Fan, Sumana Setty, Alexander Cristofalo, Sarah Pizzano, and Emi Kullberg for assistance in histology; Prakhar Sahay for developing scripts for the web site. This work was supported by the Human Frontier Science Program Long Term Fellowship to Y.S., the David and Lucile Packard Foundation to C.L., and EY022360, NS075007 and MH105949 to S.N.

## Additional information

#### Competing interests

SBN: Reviewing editor, *eLife*. The other authors declare that no competing interests exist.

## Funding

| Funder | Grant reference number | Author |
|---|---|---|
| National Eye Institute | EY022360 | Sacha B Nelson |
| National Institute of Mental Health | MH105949 | Sacha B Nelson |
| National Institute of Neurological Disorders and Stroke | NS075007 | Sacha B Nelson |
| Human Frontier Science Program | LT00151 | Yasuyuki Shima |

The funders had no role in study design, data collection and interpretation, or the decision to submit the work for publication.

## Author contributions

YS, Conception and design, Acquisition of data, Analysis and interpretation of data, Drafting or revising the article; KS, PT, Acquisition of data, Analysis and interpretation of data, Drafting or revising the article; CMH, Conception and design, Acquisition of data, Drafting or revising the article; MS, JBB, SM, Acquisition of data, Analysis and interpretation of data; CL, Conception and design, Drafting or revising the article, Contributed unpublished essential data or reagents; SBN, Conception and design, Analysis and interpretation of data, Drafting or revising the article

## Author ORCIDs

Sacha B Nelson, http://orcid.org/0000-0002-0108-8599

## Ethics

Animal experimentation: This study was performed in strict accordance with the recommendations in the Guide for the Care and Use of Laboratory Animals of the National Institutes of Health. All of the animals were handled according to approved institutional animal care and use committee (IACUC) protocols (#14004) of Brandeis University. All surgery was performed under ketamine and xylazine anesthesia, and every effort was made to minimize suffering.

# Additional files

## Supplementary files

• Supplementary file 1. Enhancer trap line data List of lines generated in this study. Line names beginning with the letter P are PiggyBac lines, others are lentiviral. Insertion sites and brief description of expression patterns are shown.

• Supplementary file 2. Annotations of line expression Expression evaluation of the PB lines.

## Major datasets

The following dataset was generated:

| Author(s) | Year | Dataset title | Dataset URL | Database, license, and accessibility information |
|---|---|---|---|---|
| Sugino K, Shima Y, Nelson S | 2016 | A Mammalian Enhancer trap Resource for Discovering and Manipulating Neuronal Cell Types | http://www.ncbi.nlm.nih.gov/geo/query/acc.cgi?token=yvmnuueudrgblwn&acc=GSE75229 | Publicly available at the NCBI Gene Expression Omnibus (Accession no: GSE75229). |

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
