## [Decision Letter]

Thank you for submitting your work entitled "A Mammalian Enhancer trap Resource for Discovering and Manipulating Neuronal Cell Types" for consideration by *eLife*. Your article has been favorably evaluated by a Senior editor and two reviewers, one of whom is a member of our Board of Reviewing Editors.

The reviewers have discussed the reviews with one another and the Reviewing Editor has drafted this decision to help you prepare a revised submission.

Summary:

Although enhancer trap has been used in other genetic model organism for transgene expression (notably *Drosophila*), it has not been widely used in mice despite previous attempts. This is mostly because previous efforts have not produced large enough scale, detailed enough characterization of expression patterns, and strong enough expression levels. Here, Shima and colleagues used the powerful tTA/TRE system and tested basal promoter, construct variants, and methods of transgene production, and produced a large collection of tTA enhancer trap lines that are expressed in highly restricted patterns in the brain. The authors have carefully characterized these expression patterns so that they can be immediately useful for many users. The authors have further followed up a few lines that led them to identify a new cell type in the piriform cortex and access a new subpopulation of thalamus-projecting layer 6 cortical neurons. The data are of high quality and the paper is well written. In addition, the authors are to be complimented for describing many technical details-including optimization procedure and failed experiments, which will be very valuable for the field. Both reviewers are enthusiastic in support the publication, assuming that the authors will address the points raised by the review below:

Essential revisions:

1) Does the restricted expression of the mCitrine reporter, which is used to characterize enhancer trap for most experiments, caused by enhancer trapping of tTA alone, or could TRE also contribute (i.e., the genomic TRE is partially silenced due to insertion site)? This is important; if the latter is the case, then when TRE transgene is expressed from a virus, the transgenes may be expressed with less specificity because the tTA pattern is actually broader than the bi-cistronic enhancer reporter (I presume that in Figure 1—figure supplement 2, tTA and TRE are independent transcription units with their own polyAs, which should be added to avoid confusion). The authors should definitely discuss this possibility, and perform additional experiments to test, using highly restricted lines such as those described in Figure 8, to validate that TRE-transgenes from AAV is expressed in the same restricted patterns as mCitrine.

2) It is unclear whether the authors have mapped their piggyBac insertions to specific genomic regions. Data in Figure 2—figure supplement 1 (local hopping) seems to suggest that authors know the insertion sites. If so, they should report all the insertion sites in a supplemental table, which will be very valuable.

3) In general, the figures can benefit from more annotations of the anatomical structures, which can help readers to appreciate the specificity and educate them about brain anatomy.

4) Please be careful in the use of "cell type".

5) It can be helpful for the authors to perform in a limited cases retrograde tracing to further define "cell type". We consider this to be optional, and the authors can determine the number of lines and which sites to perform these experiments.

6) Please unify anatomical nomenclature in the text and figures to be consistent with a single atlas (Allen Brain Atlas is recommended).

---

## [Author Response]

Essential revisions:

1) Does the restricted expression of the mCitrine reporter, which is used to characterize enhancer trap for most experiments, caused by enhancer trapping of tTA alone, or could TRE also contribute (i.e., the genomic TRE is partially silenced due to insertion site)? This is important; if the latter is the case, then when TRE transgene is expressed from a virus, the transgenes may be expressed with less specificity because the tTA pattern is actually broader than the bi-cistronic enhancer reporter (I presume that in Figure 1—figure supplement 2, tTA and TRE are independent transcription units with their own polyAs, which should be added to avoid confusion). The authors should definitely discuss this possibility, and perform additional experiments to test, using highly restricted lines such as those described in Figure 8, to validate that TRE-transgenes from AAV is expressed in the same restricted patterns as mCitrine.

To answer to this concern, we examined the possibility of TRE silencing using viral reporter injections, as the reviewers suggested. We injected TRE-myristoylated-mCherry-HA virus into several available lines and the results are shown as Figure 2—figure supplement 3. We quantified the number of infected cells at the injection sites, the numbers of mCitrine positive neurons at the injection sites, the numbers of double-labeled cells and the number of cells expressing the viral mCherry that did not express mCitrine. The infection rate was 55+/-6% (fraction of mCitrine cells at the site positive for mCherry) and the off target expression rate was 0.7% +/- 0.7% (fraction of mCherry cells negative for mCitrine). The cell counts are provided as supplemental data and described in the first paragraph of the subsection “Transgene expression in tet lines”.

The specific expression in mCitrine-labeled cells and the minimal off-target expression supports the view that specific reporter expression is due to specific tTA expression rather than TRE silencing in other cells. Occasionally, we observed a few cells without visible mCitrine expression but with viral reporter expression (arrows in Figure 2—figure supplement 3). Since those “off-target” expressing cells had similar morphologies and anatomical location to those of mCitrine positive cells, it is possible that this “off-target” expression could result from competition over tTA proteins between a single copy TRE of mCitrine in the genome and many copies of TRE from virus, but low levels of TRE silencing cannot be entirely ruled out.

We also tried two other methods to assess TRE silencing: we attempted to examine colocalization of mCitrine reporter and tTA by dual-color in situ hybridization and by immunohistochemistry. However, neither method gave high-quality tTA signals. From sequencing results, tTA expression levels (TPM) are 5 to 10% of that of mCitrine, perhaps contributing to the difficulty in detection.

(I presume that in Figure 1—figure supplement 2, tTA and TRE are independent transcription units with their own polyAs, which should be added to avoid confusion).

In all the enhancer trap probes tTA and mCitrine share polyA signals. Lentiviral constructs usually use the 3’ LTR as polyA signal and do not contain additional poly A signals because a poly A signal in the middle of the construct would terminate lentiviral RNA genome transcription and thus lower the yield of virus. We used a unidirectional polyA signal from bGH in hsp-tet3 to encode transcripts in the opposite direction, and the polyA signal did not interfere with virus production. We added an explanation of this strategy to the legend of Figure 2.

2) It is unclear whether the authors have mapped their piggyBac insertions to specific genomic regions. Data in Figure 2—figure supplement 1 (local hopping) seems to suggest that authors know the insertion sites. If so, they should report all the insertion sites in a supplemental table, which will be very valuable.

[Supplementary-material SD2-data] has all of the insertion site information. We modified the text (subsection “PB line expression patterns”, first paragraph) to state this more clearly. The insertion site information is also available via the enhancer trap website.

*3) In general, the figures can benefit from more annotations of the anatomical structures, which can help readers to appreciate the specificity and educate them about brain anatomy.*

Annotations are added to Figure 1, Figure 2, Figure 4—figure supplement 1, Figure 5, Figure 6, and Figure 7.

*4) Please be careful in the use of "cell type".*

As the reviewers correctly point out, a clear consensus on the terms and methods used to define neuronal cell types is still lacking. In response to this issue, we have tried to clarify our terminology, although we of course fall short of solving this enduring problem. Specifically, we have adopted the term “operational cell type” to refer to a population of neurons that appear to share phenotypic features and that can be defined in terms of the intersection of a transgenic strain and a brain region (see subsection “Lentivirus transgenesis”, second paragraph and Discussion, fifth paragraph). We imagine cell types organized in a tree-like taxonomy and note that operational cell types may refer to any node or branch in this tree-like structure. We distinguish this more general term from “atomic” cell types lying at the terminal branches (“leaves”) of this tree of cell types, and which are presumed indivisible in terms of their phenotypic properties. For convenience we use the term “cell type” to refer to the broader concept of “operational cell type.” A major point of the paper is to demonstrate a “proof of principle” in the case of Piriform cortex cells and layer 6 corticothalamic projection neurons, that the more specific populations labeled in our enhancer trap lines can, with further characterization, be shown to be at least closer to “atomic” cell types than previously appreciated for these populations. However, carrying this program forward is going to be a large and presumably distributed effort involving many laboratories and many approaches.

*5) It can be helpful for the authors to perform in a limited cases retrograde tracing to further define "cell type". We consider this to be optional, and the authors can determine the number of lines and which sites to perform these experiments.*

This is related to essential revision point #4. As noted above, further analyses will be needed to examine if the labeled cells are atomic cell types or can be further subdivided based not only on projections, but also based on morphology, physiology and gene expression. In some cases (e.g. LGd neurons projecting to VISp) retrograde labeling is unlikely to reveal further details, while in many other cases such experiments will help clarify whether one or two or more than two closely related atomic cell types are targeted. We feel that further experiments of this sort are beyond the scope of the current paper but they represent a major goal for future work. We have been sharing our lines with several laboratories and hope these and future collaborations will contribute to further characterization of cell types.

*6) Please unify anatomical nomenclature in the text and figures to be consistent with a single atlas (Allen Brain Atlas is recommended).*

Throughout the text we unified anatomical nomenclature to those of the Allen Brain Atlas.